

# Measurement report: Dust and anthropogenic aerosols vertical distributions over Beijing—dense aerosols gathered at the top of the mixing layer

Zhuang Wang[1,2,3], Chune Shi[1,2,3], Hao Zhang[1,2,3], Congzi Xia[9], Yujia Chen[1,2,3], Yizhi Zhu[4,5], Suyao Wang[10], Xiyuan Chi[8], Kaidi Zhang[1,2,3], Xintong Chen[1,2,3], Chengzhi Xing[4*], Cheng Liu[5,4,6,7*]

[1]Anhui Province Key Laboratory of Atmospheric Science and Satellite Remote Sensing, Anhui Institute of Meteorological Sciences, Hefei 230031, China

[2]Shouxian National Climatology Observatory, Shouxian 232200, China

[3]Huaihe River Basin Typical Farmland Ecological Meteorological Field Science Experiment Base of CMA, Shouxian 232200, China

[4]Key Lab of Environmental Optics and Technology, Anhui Institute of Optics and Fine Mechanics, Hefei Institutes of Physical Science, Chinese Academy of Sciences, Hefei 230031, China

[5]Department of Precision Machinery and Precision Instrumentation, University of Science and Technology of China, Hefei, 230026, China.

[6]Center for Excellence in Regional Atmospheric Environment, Institute of Urban Environment, Chinese Academy of Sciences, Xiamen 361021, China

[7]Key Laboratory of Precision Scientific Instrumentation of Anhui Higher Education Institutes, University of Science and Technology of China, Hefei, 230026, China.

[8]National Meteorological Center, Beijing 100081

[9]GBA Branch of Aerospace Information Research Institute, Chinese Academy of Sciences,Guangzhou 510530, China.

[10]Huaibei Meteorological Bureau, Huaibei 235000, Anhui, China

*Correspondence to*: Chengzhi Xing (xingcz@aiofm.ac.cn), Cheng Liu (chliu81@ustc.edu.cn)

**Abstract.** Over the past decades, Beijing has been suffering from persistent air pollution caused by both fine and coarse atmospheric particles. Although there are plenty of theoretical and observational studies on aerosols in Beijing, most of them only consider total aerosol concentrations and focus on heavy pollution episodes, the long–term vertical distributions of dust (coarse) and anthropogenic aerosols (fine) and their relationships with mixing layer height (MLH) have not been revealed. In this study, the dust and anthropogenic aerosols mass concentration, and MLH were retrieved by polarization Raman lidar over Beijing from May 2019 to February 2022. We found large amounts of anthropogenic aerosols accumulate at the top of the mixing layer, which is most noticeable in summer, with monthly mean mass concentration up to 57 μg/m$^3$. It is mainly influenced by the southward transport in the upper air, where the atmosphere is relatively stable and moist, favoring hygroscopic growth of particles. Dust mass concentration is discontinuous in the vertical direction. Not only on the ground but also in lofted layers that reach up to several kilometers. The heights of these lofted dust layers exhibited apparent seasonal dependence, with the height of the main dust layer gradually ascending from 1.1 km to about 2.5 km from April to June and below 3 km from October to December. In addition, there is a significant negative correlation between bottom anthropogenic



aerosols mass concentration and MLH, and an inverse function fit is more suitable to characterize this relationship, while the relationship between bottom dust mass concentration and MLH is insignificant. These results will enhance our understanding of the sophisticated interactions between dust and anthropogenic aerosols, MLH, and regional transport in northern China. It will also help to refine atmospheric chemistry models and improve surface prediction capabilities.

## 5    1 Introduction

For the past decades, atmospheric particle pollution has attracted a great deal of public concern in China, because they can decrease visibility and have substantial adverse effects on human health (Liang et al., 2022). Also, coarse and fine particles considerably affect the Earth's climate through direct and indirect effects of aerosols, and there are considerable differences in the effects of the aerosol types and particles vertical distributions on the Earth's climate (Su et al., 2020; Liu et al., 2022; Liu et al., 2021). Beijing used to be one of the most polluted cities in China, suffering from persistent atmospheric particulate pollution caused by atmospheric fine particles (anthropogenic aerosols) and coarse particles (Asian dust) (Gui et al., 2022; Zheng et al., 2015; Song et al., 2023; Huang et al., 2014; Xiao et al., 2023). Beijing is surrounded by the Yanshan and Taihang Mountains to the north and west, respectively, and the flat North China Plain (NCP) to the south, where atmospheric dispersion is relatively poor. Multiple factors contribute to the severe air pollution in Beijing, strong anthropogenic emissions due to rapid urbanization and industrialization are the crucial reasons (Su et al., 2020). In addition, the influence of Mongolia cyclone, which predispose East Asian dust to be lifted into the free troposphere (FT) and transported over long distances to the east, also affects Beijing's air quality (Gui et al., 2022; Sun et al., 2001; Xiao et al., 2023). Therefore, the observed aerosol layer in Beijing is usually a mixture of multiple types (natural and anthropogenic).

Previous studies have also shown that meteorological conditions are the key factors controlling air pollution in Beijing (Zhong et al., 2019; Guo et al., 2014; Zhong et al., 2018), with weak southerly winds, high relative humidity (RH) levels, a shallow mixing layer height (MLH), and stable atmospheric stratification associated with severe anthropogenic air pollution, while strong northwest wind is closely related to dust aerosol pollution (Wang et al., 2021c; Wang et al., 2020). Among these meteorological factors, MLH is one of the most critical, as its depth substantially determines the total volume of pollutant dispersal and mixing (Miao et al., 2018). MLH affects surface aerosol concentrations, which are associated with vertical mixing of aerosols, and can impact the dilution of surface air pollutants through various interaction and feedback mechanisms. MLH can also significantly affect the aerosol vertical distributions, as most air pollutants tend to accumulate within the mixing layer (ML) (Su et al., 2020). The development of MLH also accelerates the hygroscopic growth of fine particles within the ML, which is conductive to enhance the surface air pollution (Cheng et al., 2016; Zhang et al., 2023a; Tang et al., 2016). On the other hand, aerosols also have significant radiative feedbacks to MLH, surface dimming and upper–level heating, the magnitude and sign of the radiative feedback depends on the aerosol type and distribution height of aerosols (Huang et al., 2018; Ding et al., 2016). In a previous comprehensive review, Li et al. (2017) (Li et al., 2017) presented sufficient evidence for aerosol and MLH interactions and described their determinants. Thus, in order to more accurately estimate the formation,



growth, dissipation of air pollution and their interactions with MLH, it is necessary to observe the vertical distribution of aerosol type and its optical properties at high spatial and temporal resolution.

Although field surface observations or column measurements provide detailed optical, chemical, and microphysical characteristics of particles, they do not capture vertical distribution characteristics of aerosol types and their optical properties (Holben et al., 1998). Plenty of numerical simulation, ground based remote sensing or meteorological radiosonde have been used to acquire aerosol vertical distributions over a short time in Beijing (Zhong et al., 2019; Guo et al., 2016; Tang et al., 2016; Guo et al., 2021; Su et al., 2020), primarily focusing on severe air pollution episodes, which emphasizes the scarcity of long–term continuous measurements with high spatial and temporal resolution. Moreover, although there are more theoretical and observational studies on MLH characteristics and their correlations with surface aerosols (Zhong et al., 2019; Guo et al., 2016; Miao et al., 2015; Tao et al., 2014; Miao et al., 2018; Guo et al., 2021; Su et al., 2020; Zhong et al., 2018), most of them only consider the total aerosol concentrations, and the vertical structure and MLH relationships for coarse particles (dust) and fine particles (anthropogenic aerosols) have not been revealed. This has led to a lack of knowledge about the different aerosol types vertical distribution and their relationship with MLH. This lack of knowledge hinders the understanding of the long–range transport, mixing and stratification processes of aerosols, leading to uncertainties in the assessment of the aerosol climatic effects and meteorological chemistry models. Therefore, a comprehensive investigation of the dust and anthropogenic aerosols vertical distributions is urgently needed to facilitate mixing processes of dust with anthropogenic aerosols, and quantitative studies of the environmental contributions of different aerosol types.

The PRL (Polarization Raman lidar) technique provides a reliable tool to explore the vertical distribution characteristics of dust and anthropogenic aerosols with high spatial and temporal resolution (Tesche et al., 2009a; Ansmann et al., 2012; Tesche et al., 2017b; Tesche et al., 2009b; Tesche et al., 2017a). Moreover, the PRL is also suitable for retrieving the evolution of MLH (Sicard et al., 2006; Flamant et al., 1997). In this study, we summarize nearly 3 years of PRL observations in Beijing from May 2019 to February 2022. The annual cycle of dust and anthropogenic aerosols vertical distributions and their relationship with MLH in Beijing are clarified. The results of the PRL measurements presented here can help to fill gaps in the knowledge of the different aerosol types vertical distribution and their relationship with MLH, where limited long–term PRL observations are available. In addition, our observations can also establish a dataset of aerosol optical parameters such as particle depolarization ratio (PDR) and lidar ratio (LR) in megacities in northern China. The data can be used to support basic data analysis for spaceborne lidar missions such as CALIPSO (Cloud–Aerosol Lidar and Infrared Pathfinder Satellite Observations) (Winker et al., 2009), ALADIN (Atmospheric LAser Doppler INstrument) (Witschas et al., 2020), and ATLID (Atmospheric Lidar) (Illingworth et al., 2015), upgrading the accuracy of regional terrestrial and global satellite lidar inversions.

The objectives of our study are to (1) present a long–term dataset of aerosol LR, PDR and aerosol types in northern China, (2) elucidate the annual cycle of the dust and anthropogenic aerosols vertical distributions in Beijing, and (3) reveal the relationship between MLH and the dust and anthropogenic aerosols vertical distributions. Those results will help improve our understanding of the complex interactions between dust and anthropogenic aerosols, MLH and regional transport over northern China. It will also help refine atmospheric chemistry models and improve surface air pollution monitoring and prediction



capabilities.

## 2 Materials and methodology

### 2.1 Polarization Raman Lidar

PRL was deployed at Beijing Meteorological Observation Center (BMOC, 39.80 °N, 116.47 °E) from 22 May 2019 and

20 February 2022 (Fig.1), it is close to Beijing's South Fifth Ring, with dense traffic and high traffic flow. The PRL was placed in an air–conditioned room equipped with an uninterruptible power supply (UPS) and dehumidifier to detect aerosols over Beijing in continuous mode through a roof skylight, with a temporal resolution of half an hour, approximately 7 minutes of data collection, and 23 minutes of interval. The PRL receive system receives 532 nm perpendicular and parallel polarization backscattered signal, 355 nm Mie backscattered signal, and 387 nm nitrogen Raman backscattered signal in both analog and

photon counting modes with a spatial resolution of 7.5 m. More detailed information about the PRL system can be found in our previous study (Wang et al., 2020).

PRL can independently measure the extinction coefficient ($EXT_{355}$) and backscatter coefficient ($BAC_{355}$) at 355 nm to determine the LR at 355 nm ($LR_{355}$). Since the signal–to–noise (SNR) ratio of the Raman channel is quite low, we used a three–hour average PRL backscatter signal profile to retrieve $LR_{355}$ at night (18:00–06:00 local time) and a 12–hour average PRL

backscatter signal profile to retrieve $LR_{355}$ during the day. In addition, a four–point spatial accumulation followed by a fifteen–point spatial smoothing was applied on the average PRL backscatter signal profile. Also, an overlap function (Wandinger and Ansmann, 2002) was employed to determine the $LR_{355}$ for the lower layers. The uncertainty of the overlap correction is too high below 0.25 km, data below 0.25 km were not used for subsequent analysis. The elastic backscatter signal was used to retrieve the $EXT_{532}$ (Fernald, 1984; Klett, 1981). Fernald method requires the input of height–dependent LR and reference

heights. $LR_{355}$ is set equal to $LR_{532}$ (Sugimoto et al., 2002) to retrieve $EXT_{532}$, the closest observed $LR_{355}$ profile is used to constrain the retrieved $EXT_{532}$. According to Baars et. al (Baars et al., 2016), we use the Rayleigh fit and the five criteria in the appendix to select the optimal reference height. Volume depolarization ratio at 532 nm ($VDR_{532}$) is retrieved from the ratio of the calibrated perpendicular backscattered signal to the parallel backscattered signal at 532 nm (Freudenthaler et al., 2009; Tesche et al., 2009a), and then $PDR_{532}$ was derived from $VDR_{532}$ and the $BAC_{532}$ according to Freudenthaler et. all

(Freudenthaler et al., 2009). Also, the sampled elastic backscatter signal profiles were spatially smoothed by fifteen points to reduce the relative error in retrieving $EXT_{532}$ and $PDR_{532}$. The PRL–derived AOD is calculated by integrating the EXT profile from 0.25 to 5 km. Since the aerosols near the ground are not "seen" by the PRL due to the incomplete overlap factor. The averaged EXT between 0.25 and 0.3 km is employed to fill EXT below 0.25 km, so that AOD would not be grossly underestimated. More details on data retrieval and data validation can also be found in our previous study (Wang et al., 2020).

The relative errors of the retrieved aerosol optical parameters are estimated according to the error propagation law and primarily rely on the SNR of the average PRL backscatter signal profile (Heese et al., 2010), and we excluded signals when



the SNR was less than 1, as well as data measured under rain, snow and low cloud conditions. Percentage of analyzed PRL measurements from May 2019 to February 2022 are shown in Fig.S1a, the unanalyzed measurements of "Shutdown" and "Rain or Cloud" are due to the instrument failure or weather conditions. We filtered out data with relative errors greater than 30% for $PDR_{532}$ and greater than 60% for $LR_{355}$. For $EXT_{532}$, data with relative errors greater than 30% were filtered out when

the value was greater than 0.1 km$^{-1}$, and data with relative errors greater than 100% were filtered out when the value was less than 0.1 km$^{-1}$.

The MLH was retrieved using the gradient method (Sicard et al., 2006; Flamant et al., 1997), and a 10–point smoothing of the PRL–derived MLH was applied. All profiles were visually examined and the acquired MLH were cross–compared with the pre– and post–heights to ensure the temporal consistency of their evolution.

From January 2018 to January 2019, we conducted a study on the aerosol classification of the NCP in a farmland about 120 km north of the BMOC (Wang et al., 2021c). We used particle backward trajectories and PRL observations to summarize the aerosol optical parameters in NCP, which is mainly influenced by two aerosol types, natural dust from the northwest, prevailed in spring (March, April, May) and autumn (September, October, November), and anthropogenic aerosols from the NCP, mainly concentrated in summer (June, July, August) and winter (December, January, February). The PDR was found to

be the most relevant parameter for aerosol type, and PDR was used to classify the aerosols. When the PDR is greater than 0.23, it is classified as dust aerosol, and anthropogenic aerosols are determined by the PDR was less than 0.09. When the PDR was between 0.09 and 0.23, it is the mixture of dust and anthropogenic aerosols (also called polluted dust), and the PRL aerosol classification results are in good agreement with CALIPSO aerosol classification results. The reason PDR can be used to distinguish between pure dust and polluted dust is that dust mixes with smoke or other anthropogenic aerosols during transport,

making the mixed aerosol particles more spherical and resulting in a smaller PDR. The classification of aerosols by PDR is also widely used in other cities in China (Zhang et al., 2019b; Huang et al., 2015). Therefore, this research still uses the above PDR threshold to classify aerosols.

## 2.2 AERONET aerosol optical parameters

The sun–photometer derived aerosol optical parameters were acquired from Aerosol Robotic Network (AERONET,

https://aeronet.gsfc.nasa.gov/, last access 14 April 2023) (Holben et al., 1998), data from four sites were collected, namely "Beijing" (39.97 °N, 116.38 °E), "Beijing–CAMS" (39.93 °N, 116.32 °E), "Beijing_PKU" (39.99 °N, 116.31 °E), and "XiangHe" (39.75 °N, 116.96 °E), and the four AERONET sites around Beijing are shown in Fig. 1. In our study, AOD at 500 nm are utilized to evaluated the representativeness of PRL measurements, and coarse and fine mode AOD at 500 nm, coarse and fine mode volume concentrations are employed to convert EXT profiles to mass concentration profiles. The monthly

averaged AOD at 500 nm and conversion factor were obtained by analyzing the sun–photometer derived aerosol optical parameter around Beijing from May 2019 to February 2022.



### 2.3 Surface PM$_{10}$ and PM$_{2.5}$ mass concentration

The surface PM$_{10}$ and PM$_{2.5}$ mass concentration from 22 May 2019 and 20 February 2022 were downloaded from Beijing Municipal Ecological and Environmental Monitoring Center (http://beijingair.sinaapp.com/, last access 14 April 2023). We used the closest observation site to the BMOC (Observation number 1005A, 39.97°N, 116.47 °E), about 15 km north of the BMOC. The surface PM$_{10}$ and PM$_{2.5}$ mass concentration were screened before comparison. Firstly, outliers of PM$_{2.5}$ greater than PM$_{10}$ are discarded. Then, PM$_{10}$ and PM$_{2.5}$ mass concentration were filtered using the Z_score (Barrero et al., 2015). This was achieved by comparing each concentration to the variability of its adjacent points and time series. The filter criteria of the Z_score must satisfy three conditions at the same time: (1) the absolute Z_score is greater than 4, (2) the increment of the previous value is greater than 9, and (3) the ratio of the value to its central 3rd order moving average was greater than 2.

### 2.4 HYSPLIT Backward trajectory

The air mass backward trajectories were calculated by the Hybrid Single Particle Lagrangian Integrated Trajectory (HYSPLIT) model developed by the National Oceanic and Atmospheric Administration (NOAA) and the Global Data Assimilation System (GDAS 1º) (http://www.ready.noaa.gov, last access 14 April 2023) (Draxler and Hess, 1998). Considering the significant differences in aerosol mass concentration and types distribution at different altitudes, three trajectory arrival heights were set, 0.1 km, 2 km, and 4 km. Two–day back trajectories were generated hourly by TrajStat software, which employed HYSPLIT for trajectory calculation. Then, the backward trajectories were clustered from 2 to 4 classification directions utilizing MeteoInfo software (http://www.meteothink.org/downloads/index.html, last access 14 April 2023), detailed information can be found in (Wang et al., 2009).

### 2.5 ERA5 and MERRA reanalysis dataset

In order to filter dust and anthropogenic aerosols mass concentration profiles at high RH conditions (>85%), and to investigate the reliability of PRL derived MLH, variables such as RH, boundary layer height (BLH), etc. are needed from the ERA5 reanalysis data of the European Centre for Medium–range Weather Forecasts (ECMWF). In recent years, several studies have assessed the accuracy of the ERA5 reanalysis data provided by ECMWF based on radiosonde. For example, Luo et al. (2020) (Luo et al., 2020) found that the mean RH difference below 500 hPa between ERA5 and radiosonde was within 10%. Guo et al. (2020) (Guo et al., 2021) found that ERA5 BLH was optimal with the radiosonde BLH by comparing BLH using radiosonde data and various reanalysis data such as ERA5, MERRA–2 (The Modern–Era Retrospective Analysis for Research and Applications, Version 2), etc. The above results show that the RH and BLH of ERA5 reanalysis data have excellent accuracy and have been widely used in various studies. The ERA5 RH profile employed in this paper is divided into 15 layers below 500 hPa, with a temporal and spatial resolution of 1 h and 0.25º× 0.25º, respectively. To ensure spatial consistency, the



RH and BLH data closest to the PRL measurement site were used. In addition, the aerosol optical property data obtained from MERRA–2, including AOD, dust AOD, and black carbon AOD also employed in this study, which is provided by the National Aeronautics and Space Administration (NASA) and the Global Modeling and Assimilation Office (GMAO) (Gelaro et al., 2017).

**2.6 Polarization lidar photometer networking (POLIPHON) method**

In this study, the polarization lidar photometer networking (POLIPHON) method was applied to retrieve dust and anthropogenic aerosols mass concentration (Sugimoto et al., 2003; Shimizu, 2004; Tesche et al., 2009a; Ansmann et al., 2012; Tesche et al., 2017b; Ansmann et al., 2019). Assuming that dust and anthropogenic aerosols are externally mixed, there are two steps to obtain dust and anthropogenic aerosols mass concentration by the POLIPHON method. In the first step, the

contributions of dust and anthropogenic aerosols to the total $BAC_{532}$ are separated. Then, converting the $EXT_{532}$ of dust and anthropogenic aerosols into mass concentration (Ansmann et al., 2012; Mamouri and Ansmann, 2014). The monthly average conversion factors of dust and anthropogenic aerosols are presented in Fig. 2a. The detailed derivation of POLIPHON method and relevant input parameters involved in the POLIPHON method are summarized in Supplementary materials and Table S1, respectively.

Uncertainties in POLIPHON method are discussed in detail in previous studies (Freudenthaler et al., 2009; Tesche et al., 2017b; Mamouri and Ansmann, 2014; Tesche et al., 2009b; Tesche et al., 2017a; Mamouri et al., 2015; Wang et al., 2021b; Ansmann et al., 2011; Bravo-Aranda et al., 2015), mainly consists of two components: the input empirical parameters in Table S1 and the uncertainties of PRL derived $Bac_{532}$ and $PDR_{532}$. In general, the POLIPHON method estimates an uncertainty of 40% (Ansmann et al., 2011). BravoAranda et al. (Bravo-Aranda et al., 2015) compared the POLIPHON method directly with

field aircraft measurements and found a relative difference of less than 30%. In our measurements, the uncertainties of $Bac_{532}$ and $PDR_{532}$ below 5 km are 2%–15% and 2%–20%, respectively. The uncertainty of the retrieved mass concentration is calculated by the law of error propagation. Each input parameter in Table S1 varies within a specified range of values (here we set it as one standard deviation), while the other input parameters are set to a fixed value. The sensitivity of the parameters in Table S1 is tested one by one. Combining the uncertainty of $Bac_{532}$, $PDR_{532}$, LR, mass density and conversion factor of dust

and anthropogenic aerosols, the uncertainty of mass concentration below 5 km is between 15% and 70%.

It must be cautioned that the POLIPHON method is based on the significant difference in particle shape between the two aerosol mixtures. At higher RH, the dust and anthropogenic aerosols mass concentration separated by POLIPHON method may not be reliable. Firstly, dust and anthropogenic aerosols mass densities vary with RH, depending on the actual composition of the aerosol mixture. Secondly, the increase of RH may lead to an increase in the conversion factor. Most importantly, the

shape of the originally non–spherical particles altered accordingly at high RH. All these changes will lead to the increase of the error of POLIPHON method. Therefore, scenarios with higher RH (>85%) must be excluded when retrieving the dust and anthropogenic aerosols mass concentration (Ansmann et al., 2012; Ansmann et al., 2011). The RH profile from ERA5



reanalysis data was applied to exclude PRL derived dust and anthropogenic aerosols mass concentration profiles under high RH conditions from May 2019 to February 2022.

To ensure the reliability of the dust and anthropogenic aerosols mass concentration profiles, the PRL derived particle mass concentration at 0.25 km was compared with the surface $PM_{10}$ and $PM_{2.5}$ mass concentration. Anthropogenic aerosols mass concentration at 0.25 km was positively correlated with surface $PM_{2.5}$ mass concentration, with Pearson correlation R=0.63, P<0.01 (Fig.2). The total aerosol mass concentration at 0.25 km was compared with surface $PM_{10}$ mass concentration, and had a Pearson correlation coefficient R=0.67 (P<0.01). Due to the influence of water vapor effect (Wiegner and Gasteiger, 2015), it is difficult to get completely consistent results in the comparison. Moreover, $PM_{2.5}$ and $PM_{10}$ were measured at the ground level 15 km north of BMOC, while dust and anthropogenic aerosols mass concentration are data at an altitude of 0.25 km above the BMOC. Overall, it can be seen from the significant positive correlation in Fig.2 that POLIPHON method can reliably estimate the dust and anthropogenic aerosols mass concentration in Beijing.

## 3 Results

The data presented here were obtained in Beijing, China, by PRL, sun–photometer and reanalysis data over a period from 22 May 2019 to 20 February 2022. Since PRL data were missing from 16 November 2020 to 29 May 2021 due to laser failure, the sun–photometer observations are used to demonstrate the representativeness of PRL measurements, by statistical comparison of sun–photometer and PRL AOD. Once the representability of the PRL data has been demonstrated, the annual cycle of the AOD and MLH can be analyzed and discussed, as well as the mean vertical profile of the dust and anthropogenic aerosols retrieved by the POLIPHON method.

### 3.1 Annual cycle of AOD

Fig.3a is intended to demonstrate the representativeness of PRL measurements on a statistical basis. This is achieved by comparing the annual cycle of PRL–derived AOD and sun–photometer measured AOD, the error bar in PRL–derived AOD is calculated as one standard deviation. Fig.3a shows qualitatively and quantitatively an excellent agreement between the two datasets, with Pearson correlation and root–mean–square errors (RMSE) of 0.96 (p<0.01) and 0.018, respectively. The Angstrom Exponent was employed to convert the sun–photometer measured AOD at 500 nm to AOD at 532 nm. Although the number of PRL measurements was statistically low, the monthly AOD was similar for both data sets. This result reinforces the representativeness of PRL measurements, because it shows that the range of the PRL–derived monthly AOD set is comparable to that of the sun–photometer. The difference in the monthly mean AOD between the sun–photometer and the PRL ranges from 0.002 to 0.14, with the largest difference occurring in March, where the PRL–derived AOD is 0.14 (18%) smaller than the sun–photometer's AOD, which may be due to the absence of PRL measurements in March 2021. Two mega dust events occurred on March 2021 in Beijing, with the recorded peak AOD at 500 nm larger than 2.5 measured by sun–photometer, the



satellite data registered these two massive dust events as the most intense episode at the same time in history over the past 20 years (Gui et al., 2022).

The PRL–derived annual mean AOD is 0.50±0.29 at 532 nm, The annual variation of PRL–derived AOD showed a bimodal feature, with the first peak in March (0.61±0.29), the second peak in July (0.81±0.37), and the minimum AOD in

December (0.32±0.12), sun–photometer observations also reveal the similar annual cycle. The high loading aerosol in July and March are primarily due to long–range transport air pollutants such as Asian dust (more frequent in March) and anthropogenic aerosols (more frequent in July) (Wang et al., 2021c). Fig.1b shows the PRL–derived AOD in the ML and FT. ML AOD is obtained by integrating the EXT profile from the ground to the top of MLH, while FT AOD is obtained by integrating the EXT profile from the top of MLH to 5 km. The error bar associated with the monthly mean AOD is the one standard deviation.

There are three interesting features in Fig.3b as following:

    ● FT AOD ranges from 0.14–0.35, implying that the presence of aerosol layer above the ML all year round, and FT AOD increases significantly in March and July, which emphasizes the frequent long–range transport of dust and anthropogenic aerosols in spring and autumn.

    ● The annual cycle of the ML AOD and FT AOD is almost identical (Pearson correlation R=0.90, P<0.01),

exhibiting a bimodal shape with the primary peak (0.44±0.23 in ML and 0.35±0.26 in FT) and secondary peak (0.34±0.17 in ML and 0.28±0.20 in FT) in July and March, respectively. This is a very intriguing result because it suggests a strong systematic coupling between ML and FT, with ML AOD and FT AOD having the same source. Also note that ML AOD is consistently higher than FT AOD, with more aerosols concentrated within the ML.

    ● The annual cycle of AOD observed by PRL and sun–photometer showed the highest in summer and the lowest

in winter, while the surface observation showed the diametrically contrary conclusions (Fig.4). Due to the difference in atmospheric diffusion capacity, high surface $PM_{2.5}$ concentrations typically occur in winter and lowest in summer. This implies a high load of aerosols in the upper air of Beijing, which will be discussed in detail in Section 3.3.2. Opposite annual cycle of air pollution at the surface and in the whole layer also provides motivation for further investigation of the potential relationship between surface $PM_{2.5}$ levels and columnar AOD.

**3.2 Seasonal variability of MLH, $PDR_{532}$ and $LR_{355}$**

The MLH illustrates the relationship between intensity, duration and extent of air pollution. The maximum MLH in Beijing usually appears at 15:00 local time (LT), so this study focuses on analyzing the characteristics of MLH at 15:00 LT (Wang et al., 2021a). To investigate the reliability of PRL–derived MLH, we used the daily and monthly BLH reanalysis data at 15:00 LT from ERA5 to obtain a linear fit with the PRL–derived MLH. The daily and monthly Pearson correlation coefficient

between two sequences is 0.78 and 0.92, respectively (Fig.S2), which passed the significance test at the p = 0.01 level.

Since the MLH data retrieved by PRL are reliable, it was applied to investigate the monthly variation characteristics of MLH in Beijing from May 2019 to February 2022. Fig.5 shows the seasonal density distribution and annual cycle of the MLH.





The monthly mean MLH is highest in spring (2.11±1.04 km), lowest in winter (1.17±0.51 km), and the monthly mean MLH in summer and autumn is 1.59±0.79 km and 1.28±0.68 km, respectively. The monthly mean MLH in autumn usually ranges from 0.91 to 1.43 km, and it is rare for the MLH to be above 2 km. While the MLH in spring is widely distributed (1.24–3.30 km), the high MLH (>2 km) in spring is mainly due to the strengthening of surface wind speeds (Guo et al., 2016; Su et al.,

2018). The MLH in summer is mainly distributed between 1.04 and 1.9 km, and the intense solar radiation leads to a maximum MLH of 3.907 km in summer. The MLH in winter usually ranges from 0.81 to 1.31 km, the high MLH (>2 km) in winter usually occurs in scenarios when strong northwesterly winds are prevalent. Figure 5b shows the 5th, 25th, 50th, 75th, and 95th percentile box plots and the monthly mean MLH in Beijing from May 2019 to February 2022. The monthly mean MLH increased from January to April, and showed the first peak in April (2.26±0.94 km), followed by a decline, reaching a minimum

in July (1.27±0.41 km). The high and fluctuating MLH in March, April, and May are mainly attributed by the dynamics of strong winds, although the surface temperature was not high. Cloudy and rainy prevailed in July and August, especially in July, less surface solar radiation and lower wind speed (Guo et al., 2016; Wang et al., 2021a) depress the elevation of daytime MLH. The MLH reaches its second peak (1.57±0.81km) in September due to the strong solar radiation (Miao et al., 2015), while the height is significantly lower than the MLH in May. In November and December, the weak thermal convection and wind speed

result in relatively lower MLH.

The PDR$_{532}$ and LR$_{355}$ seasonal, ML, and FT density distribution from May 2019 to February 2022 are shown in Fig.6. The broad distributions of seasonal, ML, and FT PDR$_{532}$ and LR$_{355}$ indicate the occurrence of very diverse aerosol conditions, including complex aerosol mixtures of anthropogenic and natural aerosol types and compositions. According to Cui et al. (Cui et al., 2022), the major fine atmospheric particles sources in Beijing include residential coal, residential biomass, industrial

combustion, industrial process and vehicles emission, in addition to all local sources, long–range transport of east Asian dust and anthropogenic pollution from the NCP also contributes to the observed complexity of aerosol mixtures (Wang et al., 2016). As seen in Fig.6, the PDR$_{532}$ range from values in pure anthropogenic aerosols pollution conditions (0.0 to 0.09) to typical values in heavy dust outbreak conditions (up to 0.39 at 532 nm). Also, the value of LR$_{355}$ is between 19 sr and 96 sr and cluster around 46–61 sr. All mean values and one standard deviations of the PRL–derived PDR$_{532}$ and LR$_{355}$ are summarized in Table

1. The maximum and minimum values of PDR$_{532}$ occurred in spring (8.2±6.0) and winter (4.5±3.4), respectively, and the maximum and minimum LR$_{355}$ values occurred in winter (55.0±11.4 sr) and autumn (50.2±11.6 sr). The value of FT LR$_{355}$ in the summer is slightly larger than that in ML, while in winter, the value of ML LR$_{355}$ is considerably higher than that in FT. The value of ML LR$_{355}$ in winter and FT LR$_{355}$ in summer are about 55 sr, which is typical value for anthropogenic aerosols. Moreover, the values of ML PDR$_{532}$ in winter and FT PDR$_{532}$ in the summer are 6.1±5.8 and 3.8±2.2, respectively, which

indicate that anthropogenic aerosols accumulate in the ML in winter and FT in summer.

### 3.3 Annual cycle of aerosol vertical profiles

The POLIPHON method allows the conversion of aerosol EXT profiles into dust and anthropogenic aerosols mass



concentration profiles. Vertical profiles of the monthly mean aerosol parameters are shown in Fig.7, including $EXT_{532}$, $PDR_{532}$, POLIPHON method retrieved anthropogenic aerosols and dust mass concentration, $LR_{355}$, and fraction of dust mass concentration. These monthly mean profiles are the average of a certain number of mutually uncorrelated individual profiles, some of which may contain aerosols only in the ML, while others may contain several aerosol layers in/above the ML. The number of points averaged at different heights is shown in Fig.S1b, excluding cases where the monthly average number of points is less than 1000.

### 3.3.1 $EXT_{532}$, $PDR_{532}$, and $LR_{355}$

The $EXT_{532}$ profile exhibits a clear seasonal variation (Fig. 7a). In autumn and winter, the lowest aerosol layer is shallower and has a higher value between 0.19 and 0.32 km$^{-1}$, and then the $EXT_{532}$ drops sharply with the increase of altitude. The variation of $EXT_{532}$ profiles in autumn and winter are mainly explained by meteorological conditions, the lower height of the inversion layer (usually less than 1.5 km), and the ML is weakly developed (Fig. 5), particles are hard to transport to the upper level. The $EXT_{532}$ profile in summer is highest in the range of 0.36–0.96 km with values between 0.27–0.42 km$^{-1}$, and then $EXT_{532}$ decreased slowly with increasing altitude. In spring, due to strong wind speed (Guo et al., 2016), the aerosol was lifted to a higher height, and then gently decreased. The monthly mean $EXT_{532}$ profile shows that the highest mean $EXT_{532}$ is found in July (within the height range of 0.36–0.96 km), and the monthly mean $EXT_{532}$ in July gradually increases from 0.33 km$^{-1}$ at 0.25 km to a maximum value of 0.43 km$^{-1}$ at 0.56 km. $EXT_{532}$ was lowest in December, with a relatively superficial and thin aerosol layer gathering in the lowest layer (about 0.19 km$^{-1}$). The evolution of the monthly mean $EXT_{532}$ profile is also consistent with the previous observations (Zhang et al., 2023b). We further analyzed the top height of the aerosol layer, the top height of the aerosol layer is determined at points where the $EXT_{532}$ were below 0.01 km$^{-1}$. Similar trend was found between the top height of the aerosol layer and MLH. The extremely high aerosol layer is found in April, May, August, and September (more than 5 km) and the lowest in January (about 3.4 km), implying that the vertical diffusion capacity of aerosols in Beijing is mainly influenced by the MLH.

The $PDR_{532}$ profile exhibits considerably different seasonal variations than the $EXT_{532}$ profile (Fig. 7b). In January, February, July, August, and September, the monthly mean $PDR_{532}$ in the bottom layer (0.25 km) was extremely low (less than 0.05), indicating that the lower atmosphere of Beijing was almost unaffected by dust during this period. The monthly mean $PDR_{532}$ are highest in April and May, with values around 0.10 at 0.25 km, and the value of $PDR_{532}$ is between 0.08–0.12 in the range of 1.6 to 4.1 km. The high $PDR_{532}$ values in spring are primarily impacted by the long–range transport of Asian dust. Under the influence of the spring cold front, dust from East Asia (particularly over the Taklimakan Desert and the Gobi Desert) is easily lifted into the FT and transported eastward, affecting the air quality in Beijing (Gui et al., 2022; Miao et al., 2015).

The $LR_{355}$ provides the possibility to estimate the properties and sources of the measured aerosols (Fig. 7e). Monthly mean $LR_{355}$ is highest in January, this is because the rarely affected by dust aerosol in January in Beijing. Moreover, the building warming leads to a high emission of absorbing air pollutants in January in Beijing (Zheng et al., 2019). The MEGGA–



global reanalysis data also show a higher concentration of black carbon in Beijing in winter (Fig. 8), which is consistent with the PRL observations. The monthly mean $LR_{355}$ profile of the whole year reveals that the monthly mean $LR_{355}$ is high below 2 km and has little change with a value of 52–64 sr. However, the monthly mean $LR_{355}$ profile above 2 km varies dramatically from 35 sr (more scattering) to 67 sr (moderate absorption), indicating the complexity of airborne particles transport in Beijing, the presence of more species and relatively more aging aerosol layers.

### 3.3.2 Anthropogenic aerosols

The annual cycle of anthropogenic aerosols mass concentration retrieved by the POLIPHON method is analogous to that of $EXT_{532}$ (Fig. 7c), with the lowest anthropogenic aerosols mass concentration in December, less than 15 µg/m$^3$ in the range of 0.25 to 5 km. The monthly mean anthropogenic aerosols mass concentration in October, November and January was comparable, there was a thin aerosol layer below 1.4 km (about 23 µg/m$^3$ at 0.25 km), which is due to the weak solar radiation and low MLH in autumn and winter (Fig.5b), inhibiting the upward uplift of particles. From February to May, the MLH gradually rises due to the gradual increase in surface wind speed and solar radiation, and the distribution of aerosol layer also rises significantly, while the monthly mean anthropogenic aerosols mass concentration at 0.25 km gradually decreases, from 40 µg/m$^3$ in February to 22 µg/m$^3$ in May. In June and July, the largest monthly mean anthropogenic aerosols mass concentration and the widest aerosol distribution altitude were found, especially in July, up to 57 µg/m$^3$ in the range of 0.4 to 0.9 km. This was observed by PRL measurements and could not be identified by ground observations.

Tang et al. (Tang et al., 2015; Tang et al., 2016) found that the regional transport contributes considerably in light pollution, while local contribution is dominant in heavy pollution, and low RH and high RH typically correspond to light pollution and heavy pollution, respectively. Under low RH conditions, local emissions, regional transport and physicochemical formation processes jointly dominate the aerosol mass concentration due to the substantial influence of regional transport. For high RH, RH plays an important role in the conversion of trace gases into aerosols (Zheng et al., 2015; Cheng et al., 2016). Therefore, the increase of RH is conducive to the formation of particles in the liquid phase, multiphase reactions and hygroscopic growth processes, and the main source of aerosols in heavy pollution periods will change to local humidity–related physicochemical processes, that is, the local secondary process plays a leading role in the heavy pollution (Huang et al., 2014; Guo et al., 2014). After the key meteorological factors of air pollution in Beijing are identified, the causes of high anthropogenic aerosols mass concentration in the upper air (0.4–0.9 km) over Beijing in summer can be clarified according to the above conclusions.

Firstly, there are frequent southern transmission scenarios in the upper air over Beijing in summer. As can be seen from Fig.8, the air mass backward trajectories at 2 km and 4 km in summer is considerably different from the air mass backward trajectories in other seasons. The clustering analysis of the backward trajectories show that in spring, autumn and winter, the air masses at 2 km and 4 km are primarily from the northwest, the northwest of Beijing is mainly mountainous with a relatively clean air mass, while in the further northwest, dozens of deserts are distributed. However, in summer, 40% and 21% of the air



mass at 2 km and 4 km are transmitted from the south, respectively. The densely populated, industrial NCP is the most polluted area in China (Zhang et al., 2019a). Thus, southerly transport brings anthropogenic aerosols with high RH from the NCP into the upper atmosphere of Beijing. Secondly, the hygroscopic growth of atmospheric particles further strengthens the air pollution in the upper air over Beijing. The atmospheric particle hygroscopicity in Beijing is significantly influenced by the

source of air masses (Tang et al., 2016). When southerly air masses dominate, the polluted southerly air masses (heavy polluted and high RH) will enhance the hygroscopicity of atmospheric particles in Beijing, and as the mass fraction of inorganic salts (especially nitrate) increases, the particles undergo sufficient aging and mixing process, the particle hygroscopicity can be further increased (Cheng et al., 2016; Zhang et al., 2023a).

The monthly mean wind speed and RH obtained from the ERA5 reanalysis data further confirm the above scenario, with

higher RH and weak southerly wind in Beijing during summer (Fig. 9). Low wind speed is a prerequisite for haze episodes, and high RH is favorable for air pollution formation. In addition, Beijing is cloudy in summer (cloud fraction >50%) due to the intrusion of the western Pacific subtropical high pressure (Tang et al., 2016). When the cloud fraction increases, solar radiation and MLH will drop sharply (Fig. 5b). Such unfavorable meteorological conditions (southerly transport, high RH, low MLH and wind speed) lead to a rapid formation of atmospheric particles in the upper air over Beijing in summer. Therefore,

the high anthropogenic aerosols mass concentration in the upper air (0.4–0.9 km) over Beijing in summer is mainly caused by the growth of particle hygroscopicity under the influence of southern transport.

In fact, similar upper air pollution transmission was also found in PRL observations during the rest of the seasons in Beijing. Tao et al. (Tao et al., 2014) and our previous studies (Wang et al., 2020) have analyzed in detail the persistent haze episodes of Beijing's upper air in spring, autumn and winter. Different from summer, the air pollution in the upper air of Beijing

is dominated by dust mixed with anthropogenic aerosols in spring, autumn and winter, accompanied by the hygroscopic growth of atmospheric particles. In contrast, the upper atmosphere of Beijing is most polluted in summer, because the southern transport is more frequent (Fig.8), and the upper air of Beijing is more stable and humid (Fig.9), which is more favorable for the hygroscopic growth of atmospheric particulates.

### 3.3.3 Asian dust

Meteorological conditions in Beijing vary significantly between the seasons. Air mass transport patterns are under the influence of strong Westerlies over Beijing in spring (Kok et al., 2021; Gui et al., 2022). As early as February, there was substantial dust transport within 1.6–5 km, although it was not visible at ground (Fig.7d). From March to June, Beijing was frequently subjected to dust pollution, and the distribution of dust mass concentration at different heights was quite different. The dust mass concentration is 27, 59, 61, and 41 μg/m³ at 0.25 km from March to June, respectively. With the increase of

altitude, the dust mass concentration gradually decreased, and then increased significantly after reaching a certain altitude. Dust aerosol is mainly distributed in 2.1–2.7 km (39–46 μg/m³) in March, and more widely distributed in April (1.1–3.0 km), with higher dust mass concentration between 40 and 60 μg/m³. Compared with April, the distribution height of high dust mass



concentration in FT increased further in May and June, ranging from 2.3–3.5 km and 2.5–3.9 km, respectively. The difference in dust vertical distribution is mainly affected by meteorological factors and dust sources. Firstly, with the increasement of temperature and enhancement of convection from April to June, dust particles can be lifted higher into the air. Secondly, as two major dust sources in East Asia, Taklimakan and Gobi Desert in northwest China have some differences in their eastward

transmission paths (Sun et al., 2001). The Gobi desert dust is transported eastward in the bottom troposphere and often affects the surface air quality in Beijing, while the Taklimakan Desert dust is readily lifted to the upper troposphere and transported eastward.

In July, August and September, Beijing is rarely affected by dust aerosols (Fig.7d), with monthly mean dust mass concentration below 5 µg/m$^3$ in the range of 0.25 to 1.5 km and below 10 µg/m$^3$ in the range of 1.5 to 5 km. In October and

November, there was also an obvious dust transport episode in Beijing, but its concentration and height distribution range were considerably smaller than those from March to June. The dust mass concentration in October and November at 0.25 km is 30 and 34 µg/m$^3$, respectively, and the dust in the lower FT was concentrated in the 1–2 km, with concentrations of around 25 µg/m$^3$. In December and January, the dust mass concentration less than 5 µg/m$^3$ below 1 km, while dust mass concentration increased slightly to about 10 µg/m$^3$ within 1–2 km. The proportion of dust mass concentration in the total aerosol mass

concentration at 0.25 km showed that (Fig.7f), the highest share is in April (52%), followed by May (51%), October (38%), June (37%), November (34%), and March (29%). In the FT, the proportion of dust mass concentration in the total aerosol mass concentration is 10–60% from March to June, and 12–43% from October to November.

In general, the PRL observations show that the dust is widely distributed in height, which can be consistently extended from the surface to more than 5 km, the dust concentration is discontinuous in the vertical direction, and there is stratification.

Not only on the ground but also in lofted layers that reach up to Several kilometers. The heights of these lofted dust layers showed apparent seasonal dependence, with the height of the main dust layer gradually ascending from 1.1 km to about 2.5 km from April to June and below 3 km in autumn.

### 3.3.4 Occurrence frequency of aerosol types

Three aerosol types in Beijing have been identified according to PDR$_{532}$, Asian dust, anthropogenic aerosols, and their

mixtures (polluted dust). The means and one standard deviations of the PDR$_{532}$ and LR$_{355}$ for the three aerosol types are listed in Table 2, and the vertical occurrence frequency of the three aerosol types is shown in Fig.10. In general, anthropogenic aerosols occurred more frequently than dust and polluted dust below 5 km. In January, February, July, August, and September, the occurrence frequency of anthropogenic aerosols below 1.1 km is close to 100%, and the occurrence frequency of anthropogenic aerosols above 1.1 km decreases steeper in January and February than in July, August, and September, which is

consistent with the vertical distribution characteristics of PDR$_{532}$. From March to June and October to December, the maximum occurrence frequency of anthropogenic aerosols occurred at heights of 0.39 km (95%), 0.77 km (86%), 0.9 km (76), 0.75 km (91%), 0.65 km (91%), 0.62 km (91%), and 0.69 km (99%), respectively, none of which was at the bottom (0.25 km), indicating



that a large number of elevated aerosols gather over Beijing. It is also mentioned in Section 3.3.2, the southerly winds bring the high polluted and humid air masses from the NCP over Beijing, and under stable meteorological conditions, further hygroscopic growth of atmospheric particles leads to severe air pollution in the upper atmosphere of Beijing (Tao et al., 2014; Tang et al., 2015; Tang et al., 2016). It is worth noting that these upper layer aerosols may trigger aerosol–radiation feedback

effects, which will strengthen the surface air pollution in Beijing (Zhong et al., 2018).

The maximum occurrence frequency of dust typically occurs at 2–4 km in March to June, with a frequency of 0–9% (Fig.10), and a small amount of dust aerosols also appear in other months, usually above 1 km, with a frequency of less than 3%. The vertical occurrence frequency of polluted dust also shows considerably seasonal variation. From March to June, polluted dust occurred more frequently (5–39%) below 4 km, and the height of maximum occurrence frequency was about 2.5

km. In addition, the occurrence frequency of polluted dust at 0.25 km is between 10 and 39% from March to June, indicating the considerable contribution of dust aerosol to the bottom air pollution in Beijing. From July to September, a small amount of polluted dust occurs above 1.2 km with a frequency of 0–19%. In October and November, polluted dust is mainly distributed in 0–3 km, and contributed more frequently to the bottom layer (0.25 km) and 1.2–2.4 km, up to 30%. In December, January and February, the polluted dust usually appeared in 1.1–2.4 km, while the anthropogenic aerosols concentrated below 1 km,

and aerosols tend to be vertically stratified in winter, the stratification of aerosols is mainly caused by the difference of meteorological conditions above (dominated by northwest winds) and below (dominated by southerly winds) the ML, it has been discussed in detail in our previous studies (Wang et al., 2020).

## 4 Discussions

The vertical structure of ML is critical to surface air pollution because it affects the volume of air pollutants mixing

(Miao et al., 2018). If the factor driving the variation of surface air pollution with MLH is meteorological, then the discuss of their relationship can elucidate meteorological effects. We examine the MLH and the bottom (0.25 km) dust and anthropogenic aerosols mass concentration retrieved by PRL on a case–by–case basis. Scatter plots of MLH with bottom total, anthropogenic aerosols, and dust mass concentration are shown in Fig.11. Although total aerosol mass concentration is negatively correlated with MLH, there are substantial differences and spreads in the correlations between dust and anthropogenic aerosols mass

concentration and MLH. Linear regression and inverse function fitting methods were applied to characterize the relationship between MLH and dust/anthropogenic aerosols mass concentration. Anthropogenic aerosols mass concentration and MLH were significantly negatively correlated with a Pearson correlation of -0.35. In addition, the nonlinear inverse function was in high agreement with the mean of each bin and characterized the relationship between MLH and anthropogenic aerosols mass concentration with a higher correlation coefficient (-0.46). The correlation coefficient of the inverse function fitting was greater

than the Pearson correlation coefficient, suggesting that the nonlinear fit may be better suited to characterize the relationship between MLH and bottom particles mass concentration, and this improvement was significant for total and anthropogenic aerosols mass concentration.





Anthropogenic aerosols mass concentration and MLH are significantly negatively correlated, while dust mass concentration and MLH are weakly correlated and not statistically significant (Fig.11). In fact, surface air pollution is affected by the combined effects of local emissions and meteorological conditions with different spatial and temporal distributions, and MLH is only one of these influencing factors. Previous studies (Zhong et al., 2018) also pointed out that anthropogenic air

pollution episodes in Beijing are typically divided into two processes, transmission and accumulation. In the transmission stage, air pollution is primarily induced by southerly transport, while in the accumulation stage, meteorological conditions are stagnant and anthropogenic aerosols usually exhibit explosive growth. Subsequently, the anthropogenic aerosols within the ML scatter more solar radiation back to space, thus increasing the temperature inversion and lowering the MLH, further deteriorating the surface air quality. There is a positive feedback mechanism between MLH and surface anthropogenic aerosols

in this process. Therefore, MLH shows a significant negative correlation with bottom anthropogenic aerosols mass concentration. However, dust aerosol in Beijing is governed by regional transport, and MLH is not the key parameter controlling surface dust mass concentration. Our previous study (Wang et al., 2021c) shows that, for pure dust episodes, air masses are transported from the northwest with stronger wind speed and high MLH. For polluted dust episodes, air masses are transported from northwest to the NCP and then transmitted to Beijing by southerly winds, and the generation rate of new

particles is further accelerated by the mixing of dust and anthropogenic aerosols (Nie et al., 2014), which is prone to persistent heavy haze events with low MLH at this time. Therefore, the bottom dust mass concentration is mainly influenced by transport, and the correlation between bottom dust mass concentration and MLH is insignificant in long–term statistics.

The analysis in sections 3.1 and 3.2 shows that since both MLH and air pollution in Beijing undergo the most pronounced seasonality, seasonal differences between MLH and aerosol are further analyzed. Fig.12 focuses on the seasonal dependence

of MLH and bottom dust and anthropogenic aerosols mass concentration. Similar to Figure 11, there is a significant negative correlation between anthropogenic aerosols and MLH in four seasons, while dust particles and MLH are weakly correlated. The maximum magnitude of slope between -1/MLH and anthropogenic aerosols mass concentration is about 46 km μg/m$^3$ in summer, followed by 38 km μg/m$^3$ in winter, the spring and autumn are comparable, with values of 26 and 28 km μg/m$^3$, respectively. It indicates that the sensitivity of anthropogenic aerosols to MLH is strongest in summer, followed by winter and

weakest in spring and autumn. The anthropogenic aerosols do not increase linearly with the decrease of MLH. When MLH was less than 1.2 km, anthropogenic aerosols increased rapidly with the decrease of MLH, while anthropogenic aerosols changed more slowly when the MLH was greater than 1.2 km, and the relationship between MLH and anthropogenic aerosols is virtually flat when MLH greater than 3 km.

In addition, Fig.13 shows the total, anthropogenic aerosols, and dust mass concentration profiles as a function of MLH,

as derived by PRL in Beijing from May 2019 to February 2022, with the POLIPHON method applied. Fig.14 shows the seasonal relationship between PRL–derived MLH and vertical profiles of total, non–dust, and dust mass concentration. The interval of MLH is 0.2 km, we require that the number of averaged particle mass concentration profiles and MLH samples larger than 20 for each interval of MLH, otherwise, the data will be discarded, as shown in white area in Fig.14.

The anthropogenic aerosols mass concentration within the ML decreases continuously with increasing ML (Fig. 13).





When MLH is less than 0.4 km, high total aerosol mass concentration (>50 μg/m³) extends up to 1.6 km, in which the anthropogenic aerosols (about 60 μg/m³) are primarily in ML, and the dust aerosol (about 50 μg/m³) is mainly above the ML, and this aerosol stratification at low MLH usually occurs in winter (Fig.14j–l). When the MLH is between 0.4 and 2.0 km, the anthropogenic aerosols mass concentration increased from the bottom to the top of ML, and reaches a maximum at the top of the ML (35–60 μg/m³). The accumulation of anthropogenic aerosols at the top of the ML is most significant in summer, with concentrations up to 57 μg/m³, and similar phenomenon also exists in spring, autumn and winter, but the aerosol mass concentration is lower. We have discussed this phenomenon in detail in Section 3.3.2, because the southward transport is more frequent in summer (Fig.8) and the air pollutants aloft in Beijing are more stable and moist (Fig.9), which is more favorable for the hygroscopic growth of atmospheric particles at the top of MLH. When the MLH is greater than 2 km, the anthropogenic aerosols mass concentration from the bottom to high altitude is less than 15 μg/m³, and the atmosphere is relatively clean.

The relationship between MLH and anthropogenic aerosols vertical profiles is quite different from the relationship between MLH and dust vertical profiles. No matter how much MLH is, there is always a shallow dust layer in the bottom layer, with mass concentration between 8 and 40 μg/m³ (Fig.13c), which further confirms that there is no significant correlation between MLH and the bottom dust mass concentration. Moreover, high dust mass concentration is also distributed near the ML, with maximum mass concentration up to 104 μg/m³. Seasonal variations show that (Fig.14), when the MLH ranges from 0.8 to 1.6 km in spring, high dust mass concentration extends up to 3.2 km, while dust aerosols are mainly concentrated near the ML when the MLH is greater than 1.6 km. Different from spring, dust mass concentration is relatively low when ML is less than 1.6 km in summer, while high dust mass concentration is found around ML. The dust mass concentration in the ML is slightly lower than that above the ML in autumn and winter when the ML is below 1.6 km.

In general, there is a significant negative correlation between bottom anthropogenic aerosols and MLH. When MLH is between 0.4 and 2 km, anthropogenic aerosols typically accumulate at the top of ML, and the bottom anthropogenic aerosols have lower concentrations, especially in summer. Aerosols are prone to stratification when the MLH is less than 0.4 km. The relationship between dust and MLH is complicated. There is no clear relationship between bottom dust mass concentration and MLH, while the dust in the upper air tends to be distributed near ML.

## 5 Conclusions

PRL was deployed to investigate the aerosol vertical distributions and their relationship with MLH over Beijing from 22 May 2019 and 20 February 2022. The broad distributions of seasonal, ML, and FT PDR$_{532}$ and LR$_{355}$ were found over Beijing, indicating the occurrence of very diverse aerosol conditions. The PDR$_{532}$ ranges from values in pure anthropogenic aerosol pollution conditions (0.0 to 0.09) to typical values in heavy dust outbreak episodes (up to 0.39). Also, the LR$_{355}$ is between 19–96 sr and clusters around 46–61 sr. The aerosol classification results also show that, besides anthropogenic aerosols, the maximum occurrence frequency of dust usually occurs at 2–4 km (0–9%) in March to June, and the maximum occurrence frequency of polluted dust usually appeared in 2.5 km (13–39%) from February to June and 1.5 km (5–30%) from November





to January. These valuable datasets acquired through PRL can be used to improve lidar algorithms for current and/or future satellite lidar missions.

The annual variation of PRL–derived AOD showed a bimodal characteristic, with the first peak in March (0.61±0.29), the second peak in July (0.81±0.37). In addition, the annual cycle of the ML AOD and FT AOD is virtually identical (Pearson correlation R=0.90, P<0.01), it implies a strong systematic coupling between ML and FT. Most importantly, the annual cycle of AOD observed by PRL and sun–photometer showed the highest in summer and the lowest in winter, while the surface observation showed the diametrically contrary conclusions. This implies a high load of aerosols in the upper air of Beijing. Further retrieval of dust and anthropogenic aerosols mass concentration by the POLIPHON method revealed that, in June and July, the largest monthly mean anthropogenic aerosols mass concentration and the widest aerosol distribution altitude were found, especially in July, up to 57 μg/m$^3$ in the range of 0.4 to 0.9 km. The high anthropogenic aerosols mass concentration in the upper air (0.4–0.9 km) over Beijing in summer is mainly caused by the growth of particle hygroscopicity under the influence of southern transport. The PRL observations show that the dust can be consistently extended from the surface to more than 5 km, and the dust concentration is discontinuous in the vertical direction. Not only on the ground but also in lofted layers that reach up to several kilometers. The heights of these lofted dust layers exhibited apparent seasonal dependence, with the height of the main dust layer gradually rising from 1.1 km to about 2.5 km from April to June and below 3 km in autumn and winter.

There is a significant negative correlation between the bottom anthropogenic aerosols mass concentration and MLH (R=-0.35, P<0.01) due to the positive feedback mechanism between MLH and surface anthropogenic aerosols, and an inverse function fit is more suitable to characterize this relationship (R=-0.46, P<0.01). However, there is a weak correlation between the bottom dust mass concentration and MLH in long–term statistics, which is not statistically significant, because the bottom dust mass concentration is mainly influenced by transport. The relationship between MLH and anthropogenic aerosols mass concentration profiles shows that, anthropogenic aerosols mass concentration increase from the bottom to the top of the ML and reach a maximum at the top of the ML (35–60 μg/m$^3$). The accumulation of anthropogenic aerosols at the top of the ML is most pronounced in summer, with monthly mean mass concentration up to 68 μg/m$^3$, and similar phenomena also exist in spring, autumn and winter, but with lower mass concentration. This is because the upper air southward transport is more frequent in Beijing in summer, and the atmosphere is more stable and moist, which is more conducive to the hygroscopic growth of atmospheric particles.

The research on dust and anthropogenic aerosols vertical distributions and their correlation with MLH contribute to further understanding of the contribution of dust/anthropogenic aerosols to the environment and climate, and helps to supplement the dust/anthropogenic aerosol database with useful empirical information for upgrading atmospheric chemistry models and air pollution forecasting and warning.

*Author contributions.* Cheng Liu and Chengzhi Xing conceived and supervised the study; Zhuang Wang analyzed the Raman Lidar data; Zhuang Wang wrote the manuscript with input from Cheng Liu, Chengzhi Xing Chune Shi, and Hao Zhang



reviewed and commented on the paper; All authors contributed to discuss the results and revised manuscript.

*Competing interests.* The authors declare that they have no competing interests.

*Code/Data availability.* Measurement data in this study are available in the data repository maintained by Mendeley Data
https://doi.org/10.17632/7k7czvw7ty.1 (Wang et al., 2023). Additional code/data related to this paper may be requested from
the authors.

*Acknowledgments.* The authors acknowledge the National Oceanic and Atmospheric Administration (NOAA) Air Resources
Laboratory (ARL) for the provision of the HYSPLIT transport and dispersion model used in this publication. We thank all the
researchers for their efforts in establishing and maintaining the Beijing AERONET site. We would like to thank the Beijing
Ecological Environment Monitoring Center for the $PM_{2.5}$ and $PM_{10}$ data. We would like to thank the ERA5 and MERRA-2
data developers for providing free and open source reanalysis materials. This research was supported by grants from the Anhui
Provincial Natural Science Foundation "Jianghuai Meteorological" Joint Fund (2208085UQ04), National Natural Science
Foundation of China (No. 42005095), the Joint Research Project for Meteorological Capacity Improvement(22NLTSQ011,
22NLTSY006),

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



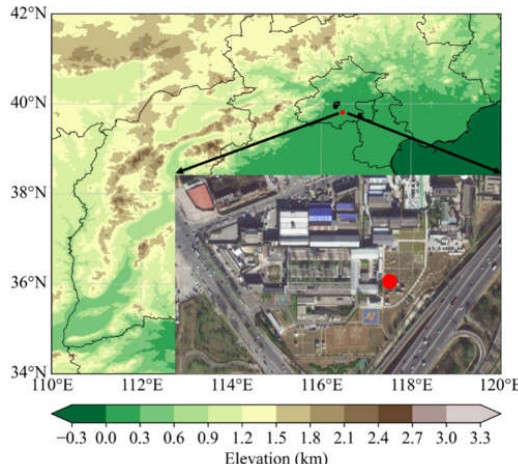

**Fig.1**. Topographic map of Beijing Meteorological Observation Center (BMOC). The red dot marks the position of PRL observation station, the black dot marks the sun–photometer position, and the lower–right image is the true color image of the BMOC (the true color image of BMOC is available at https://ditu.amap.com/, last access 14 June 2023).



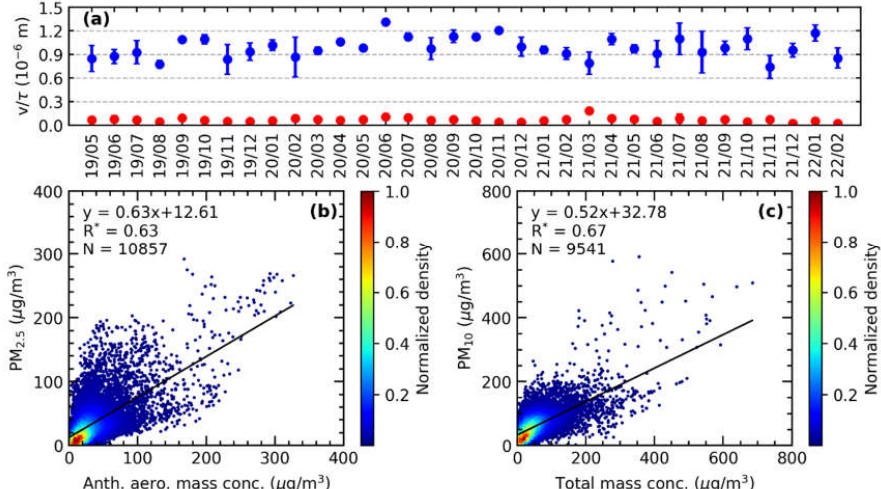

**Fig.2**. Sun–photometer observed (**a**) monthly averaged anthropogenic aerosols (blue) and dust (red) conversion factor from May 2019 to February 2022. Scatter plots show the comparison between (**b**) surface $PM_{2.5}$ mass concentration and PRL–derived bottom (0.25 km) anthropogenic aerosols mass concentration, (**c**) surface $PM_{10}$ mass concentration and PRL–derived bottom total mass concentration. The black line is the linear fitting line, and the correlation coefficients are shown at the top left of (**b**) and (**c**), N = number of samples, the asterisk on correlation coefficients R stands for P<0.01.



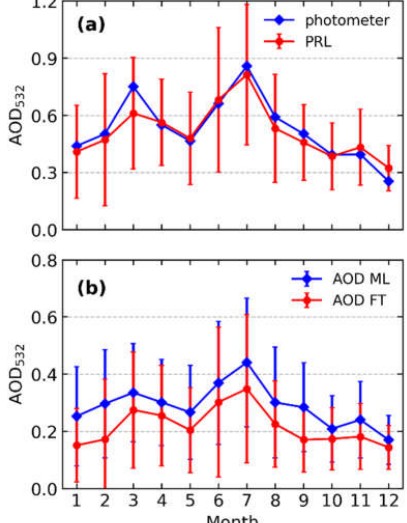

**Fig.3**. Annual cycle of the AOD at 532 nm: (**a**) PRL–derived and sun–photometer measured AOD, (**b**) PRL–derived AOD in the mixing layer (ML) and free troposphere (FT) . The envelope over the vertical bars represent one standard deviation.

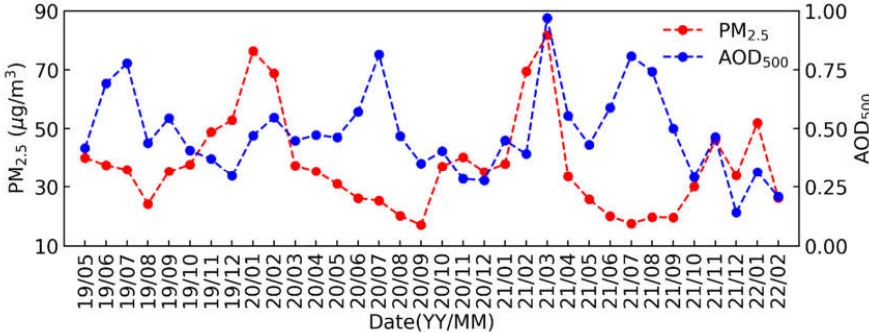

10    **Fig.4**. Surface monthly mean $PM_{2.5}$ mass concentration collected from Beijing Municipal Ecological and Environmental Monitoring Center and AOD at 500 nm measured by sun–photometer from May 2019 to February 2022.



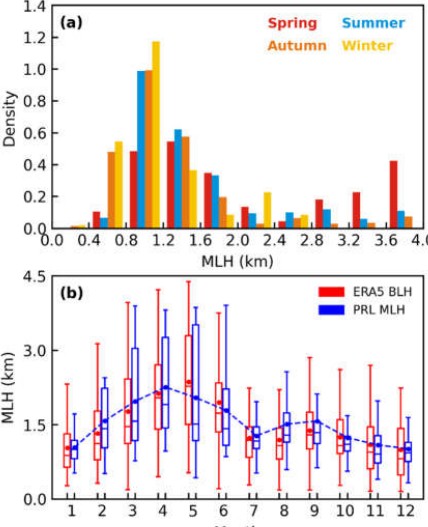

**Fig.5**. Seasonal density distribution of (**a**) PRL–derived MLH, and (**b**) annual cycle of PRL–derived MLH and BLH in ERA5 reanalysis data. Note that the MLH and BLH are the monthly mean value at 15:00 LT. The box and whisker plots showing the $5^{th}$, $25^{th}$, $50^{th}$, $75^{th}$, and $95^{th}$ percentiles, the dots represent the monthly mean MLH.

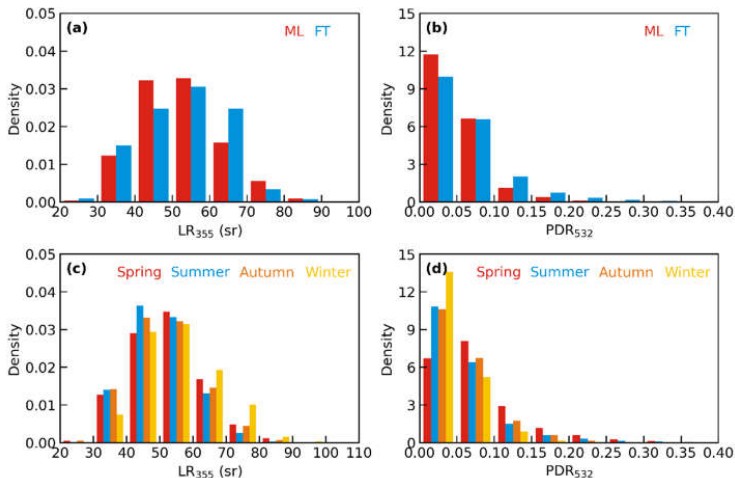

**Fig.6**. Density distribution of PRL–derived aerosol optical parameters: (**a**) ML $LR_{355}$ and FT $LR_{355}$, (**b**) ML $PDR_{532}$ and FT $PDR_{532}$. Seasonal density distribution of (**c**) $LR_{355}$ and (**d**) $PDR_{532}$.



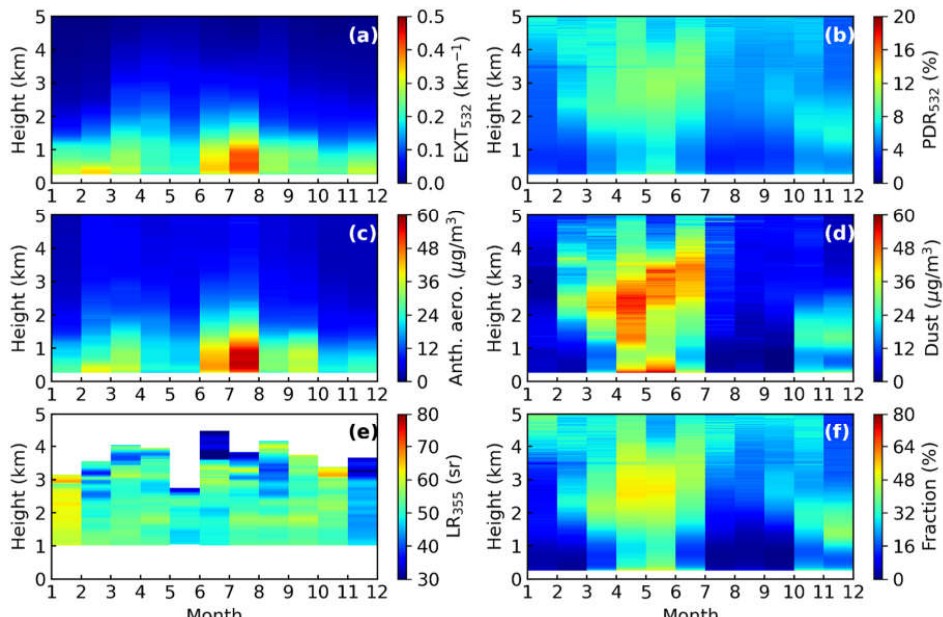

**Fig.7**. Vertical profiles of the monthly mean aerosol optical parameters: (**a**) $EXT_{532}$, (**b**) $PDR_{532}$ (**c**) anthropogenic aerosols mass concentration, (**d**) dust mass concentration, (**e**) $LR_{355}$, and (**f**) fraction of dust mass concentration in total aerosol mass concentration in Beijing from May 2019 to February 2022.



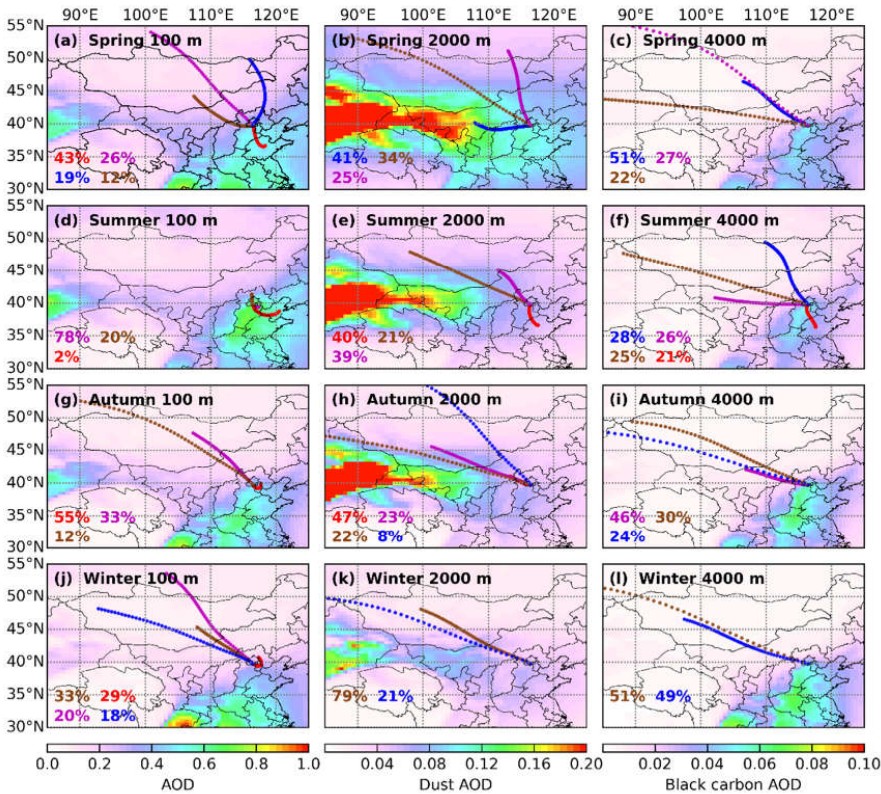

**Fig.8**. Cluster analysis of seasonal 48–hour air mass backward trajectories in Beijing from May 2019 to February 2022: initialized at (**a**) 100 m, (**b**) 2 km, and (**c**) 4 km in spring, initialized at (**d**) 100 m, (**e**) 2 km, and (**f**) 4 km in summer, initialized at (**g**) 100 m, (**h**) 2 km, and (**i**) 4 km in autumn, and initialized at (**j**) 100 m, (**k**) 2 km, and (**l**) 4 km in winter. We calculated the hourly air mass backward trajectories during each season. Then, cluster analysis was carried out in 2–4 categories directions. The percentages at the bottom right of each subplot indicate the percentage of each backward trajectory. The color on each subplot indicates the AOD, dust AOD, and black carbon AOD for each season obtained from the MEGGA–2 global reanalysis data.



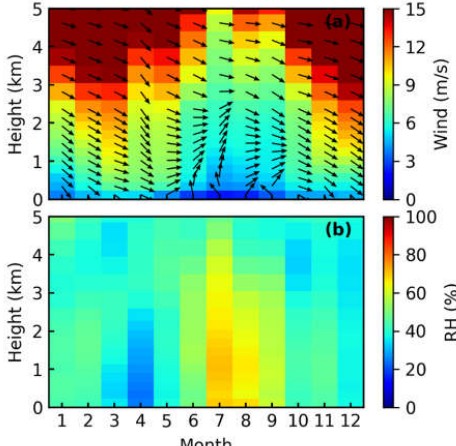

**Fig.9**. Vertical profiles of the monthly mean (**a**) wind speed and direction, (**b**) RH obtained from the ERA5 reanalysis data from May 2019 to February 2022. The black arrow in (**a**) shows the wind direction, and the upward indicates the south wind.

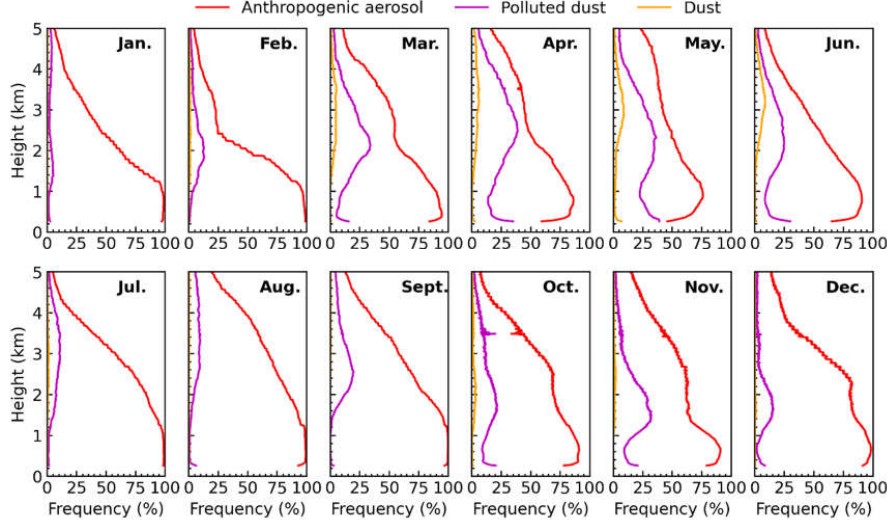

**Fig.10**. Vertical distributions of the monthly mean occurrence frequencies of dust (yellow), polluted dust (magenta), and anthropogenic aerosols (red) in Beijing from May 2019 to February 2022.





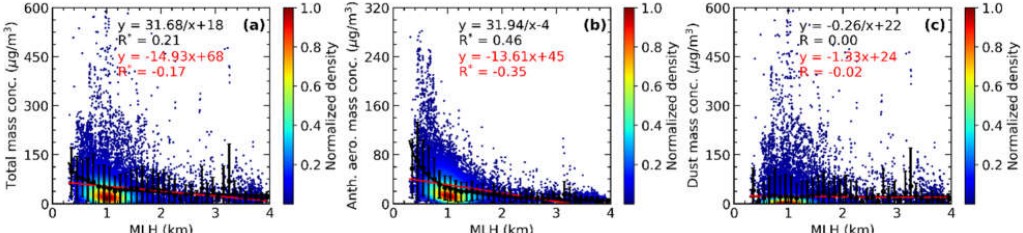

**Fig.11**. The relationship between PRL–derived MLH and (**a**) total aerosol mass concentration, (**b**) anthropogenic aerosols mass concentration, and (**c**) dust mass concentration at 0.25 km in Beijing from May 2019 to February 2022. The black dots and vertical bars represent the mean values and one standard deviation for each bin. The interval of each bin is 0.2 km. The black and red lines represent the linear fit ($f(x) = Ax + B$) and inverse fit ($f(x) = A/x + B$), respectively. The fitting functions and correlation coefficients for linear fit (black) and inverse fit (red) are shown in top right of each subplot. The asterisks on the correlation coefficient R indicate statistically significant (P < 0.01). Color shaded points represent normalized sample density.





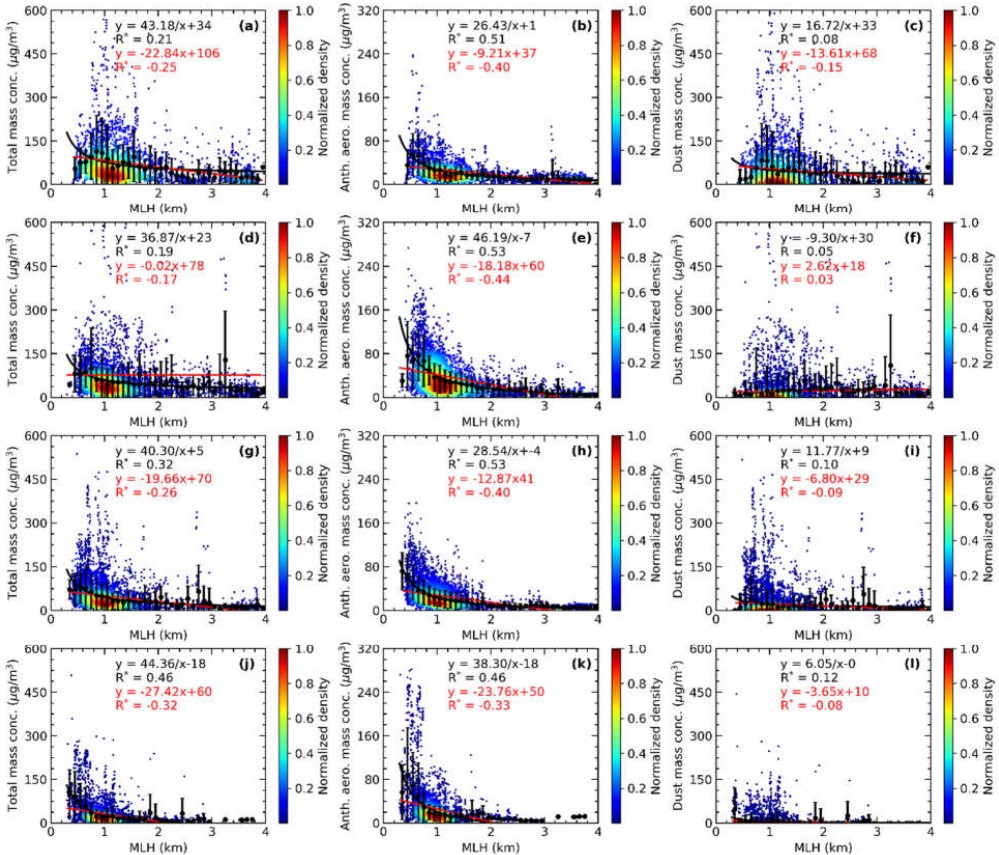

**Fig.12**. The relationship between PRL–derived MLH and total, anthropogenic aerosols, and dust mass concentration at 0.25 km in (**a–c**) spring, (**d–f**) summer, (**g–i**) autumn, and (**j–l**) winter in Beijing from May 2019 to February 2022. The black dots and vertical bars represent the mean values and one standard deviation for each bin. The interval of each bin is 0.2 km. The black and red lines represent the linear fit (*f(x) = Ax + B*) and inverse fit (*f(x) = A/x + B*), respectively. The fitting functions and correlation coefficients for linear fit (black) and inverse fit (res) are shown in top right of each subgraph. The asterisks on the correlation coefficient R indicate statistically significant (P < 0.01). Color shaded points represent normalized sample density.



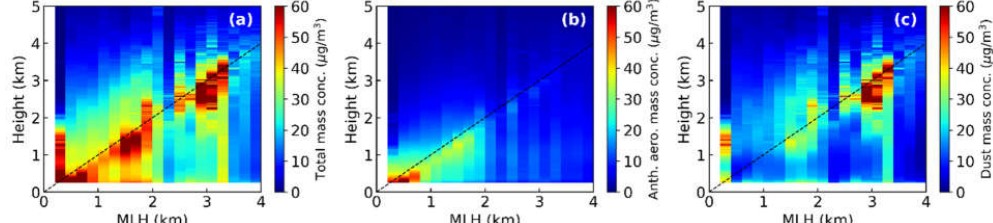

**Fig.13**. The relationship between PRL–derived MLH and vertical profiles of (**a**) total aerosol mass concentration, (**b**) anthropogenic aerosol mass concentration, and (**c**) dust mass concentration for each bin in Beijing from May 2019 to February 2022. The interval of each bin is 0.2 km. The black dash line represents the MLH.



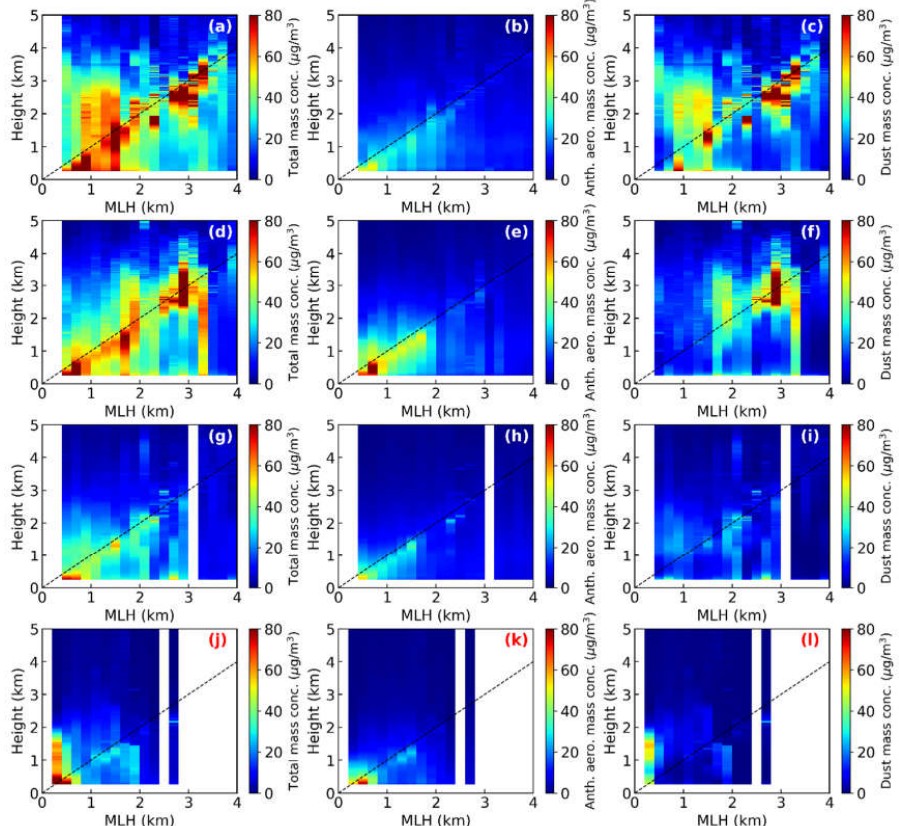

**Fig.14.** The relationship between PRL–derived MLH and vertical profiles of total, anthropogenic aerosols, and dust mass concentration in (**a–c**) spring, (**d–f**) summer, (**g–i**) autumn, and (**j–l**) winter for each bin in Beijing from May 2019 to February 2022. The interval of each bin is 0.2 km. The black dash line represents the MLH.





**Table 1**. Annual and seasonal means and one standard deviation (Calculate with ML estimation) of the $PDR_{532}$ and $LR_{355}$.

| | $PDR_{532}$ (%) | $LR_{355}$ (sr) | $PDR_{532}$ (%) (ML) | $PDR_{532}$ (%) (FT) | $LR_{355}$ (sr) (ML) | $LR_{355}$ (sr) (FT) |
|---|---|---|---|---|---|---|
| Year | 6.1±5.0 | 51.6±11.3 | 5.2±3.6 | 6.5±5.6 | 51.6±11.3 | 52.0±11.4 |
| Spring | 8.2±6.0 | 51.6±10.8 | 7.0±4.7 | 8.8±6.6 | 51.6±10.8 | 51.2±8.4 |
| Summer | 6.0±5.2 | 50.4±10.3 | 5.0±3.6 | 6.1±5.8 | 50.3±10.3 | 56.7±11.5 |
| Autumn | 5.9±4.8 | 50.2±11.6 | 5.1±3.1 | 6.2±5.4 | 50.2±11.6 | 53.2±10.2 |
| Winter | 4.5±3.4 | 55.0±11.4 | 3.8±2.2 | 4.9±3.8 | 55.2±11.4 | 46.9±12.0 |

**Table 2**. Means and one standard deviation of the $PDR_{532}$ and $LR_{355}$ for the three aerosol types.

| | Anth. Aero. | Polluted dust | Asian dust |
|---|---|---|---|
| $PDR_{532}$ (%) | 4.3±2.1 | 13.2±3.5 | 28.7±4.3 |
| $LR_{355}$ (sr) | 52.1±11.3 | 47.8±11.0 | 45.7±5.1 |

