# Peer review of "Measurement report: Dust and anthropogenic aerosols vertical distributions over northern China—dense aerosols gathered at the top of the mixing layer"

_EGUsphere, 2023_

## Author Response (AR1)

**Response to reviewer 1:**

We really appreciate the reviewers for the valuable and constructive comments, which are very useful for the improvement of the manuscript. We have replied the reviewers' comments point-to-point in below. The reviewers' comments are cited in black, while the responses are in blue. The revised parts in the manuscript are marked in red. All the page number and line number are referred to the revised manuscript.

**Major issue**

(1)  The authors speculate too much in the data analysis. For example, in page 18, line 10-12, "The high anthropogenic aerosols mass concentration in the upper air (0.4.0.9 km) over Beijing in summer is mainly caused by the growth of particle hygroscopicity under the influence of southern transport.", this statement cannot be supported by the results presented in the manuscript. As the authors described, all profiles with high relative humidity (> 85%) have been ruled out from the data analysis (see page 6, line 20-21). Then, hygroscopic growth should not be significant. And more importantly, no results of hygroscopic growth factors were shown, how this conclusion can be made? Similar issue is also laid in the analysis of MBL height and aerosol mass concentration (There is a minor issue associated with this). I hope the authors can focus on their own results and start from these results to re-think what they can conclude.

R: Thank you for your comment, which is very meaningful and valuable in improving the quality of our manuscripts. Due to the lack of measurement of the hygroscopic growth factors at the top of the mixing layer, we have no direct evidence to prove the effect of hygroscopic growth on upper anthropogenic aerosols. However, during our observation period, there are frequent southern transport scenarios in the upper air over Beijing in summer, with weak southerly winds, high relative humidity levels, and weak wind speed (Figure R1, Figure R2), the meteorological conditions are conducive to the hygroscopic growth of particles based on previous researches (Tang et al., 2015; Tang et al., 2016). Due to the limitations of the POLIPHON method, we excluded cases with relative humidity greater than 85%. However, previous studies revealed that the hygroscopic growth of particles also occurs when the relative humidity is between 40%

and 86% (Wu et al., 2020; Xia et al., 2019). The hygroscopic growth of particles at the top of the mixing layer was inferred based on previous findings and air pollution characteristics during our observation period.

We further analyzed the relationship between the relative humidity and $EXT_{532}$ and $PDR_{532}$ at the top of mixing layer in summer (Figure R3). When the relative humidity was greater than 40%, the $EXT_{532}$ increased with the increase of relative humidity, that is, the air pollution increased with the increase of humidity. More importantly, with the increase of relative humidity, the $PDR_{532}$ gradually decreases, indicating that there is hygroscopic growth of particles at the top of mixing layer (Dawson et al., 2020). Because the $PDR_{532}$ is related to hygroscopicity, it is inversely proportional to the sphericity of atmospheric particles. Assuming a uniform refractive index, moistened aerosol particles are thought to be more spherical due to condensation of water vapor and surface tension, resulting in a lower $PDR_{532}$. Thus, there is hygroscopic growth of particles at the top of mixing layer, but the contribution of hygroscopic growth to anthropogenic aerosols mass concentration at the top of mixing layer is still uncertain.

To sum up, we corrected our statement in the manuscript: "The high anthropogenic aerosols mass concentration in the upper air (0.4-0.9 km) over Beijing in summer is mainly caused by the southward transport in the upper air, where the atmosphere is relatively stable and moist, favoring hygroscopic growth of particles." This is also a limitation of our observations, which we highlight at the conclusion of the manuscript.

We also include these important results in the manuscript. Please refer to Page 13 Line 22–29, Page 17 Line 29–31, Page 18 Line 29–30, Page 19 Line 16–26, and Figure 9 in the manuscript.

[Figure]

***Figure R1***. *Cluster analysis of seasonal 48–hour air mass backward trajectories in Beijing from May 2019 to February 2022: initialized at (a) 100 m, (b) 2 km, and (c) 4 km in spring, initialized at (d) 100 m, (e) 2 km, and (f) 4 km in summer, initialized at (g) 100 m, (h) 2 km, and (i) 4 km in autumn, and initialized at (j) 100 m, (k) 2 km, and (l) 4 km in winter. We calculated the hourly air mass backward trajectories during each season. Then, cluster analysis was carried out in 2–4 categories directions. The percentages at the bottom right of each subplot indicate the percentage of each backward trajectory. The color on each subplot indicates the AOD, dust AOD, and black carbon AOD for each season obtained from the MERRA–2 global reanalysis data.*

[Figure]

*Figure R2*. *Vertical profiles of the monthly mean (**a**) wind speed and direction, (**b**) RH obtained from the ERA5 reanalysis data from May 2019 to February 2022. The black arrow in (**a**) shows the wind direction, and the upward indicates the south wind.*

[Figure]

*Figure R3*. *The box and whisker plots of the relationship between the (a) RH and $EXT_{532}$, and (b) RH and $PDR_{532}$ at the top of mixing layer in summer. The box and whisker plots showing the 5th, 25th, 50th, 75th, and 95th percentiles, the red dots represent the mean values.*

**Minor issues**

(1)   Page 5, line 8, the authors should be specific that the gradient of which quantity they used in the MBL height determination. And how do they treat the very shallow nonctual boundary layer height within the incomplete overlap region?

R: Thanks for your comments. The mixing layer height (MLH) was retrieved using the gradient method (Sicard et al., 2006; Flamant et al., 1997), which is the most classic and widely used method, gives the MLH as the altitude of the minimum gradient of the range-squared-corrected signal:

$$MLH = \min(\frac{(P(R)*R^2)}{dR}) \tag{1}$$

*P(R)* represent the backscatter signal collected by telescope from range *R*. The retrieve range of the gradient method is 0.25–4 km, so the minimum MLH is 0.25 km and the maximum MLH is 4 km. Therefore, we cannot capture the very low MLH, especially at night, which may lead to overestimation of the MLH in Beijing. This is also one of the shortcomings of PRL retrieval of MLH, and we also include these statements in the manuscript and highlight this shortcoming in conclusion. Please refer to Page 5 Line 16–20 and Page 19 Line 16–26 in the manuscript.

(2)   Page 6, line 21-22, the authors need to explain why they intended to use lidar-derived MBL height in MBL AOD calculation instead of using ERA-5 MBL height, although they think ERA-5 height is reliable and can be used to evaluate the lidar-derived MBL height.

R: The nighttime boundary layer height (BLH) of ERA5 is extremely low, only tens of meters, while the lowest detection height of PRL is 0.25 km. If the BLH of ERA5 is used, a large amount of data at night will not be available when discussing the BLH and anthropogenic aerosols/dust mass concentration, so the PRL retrieved MLH was employed. We have also added these notes to the manuscript. Please refer to Page 10 Line 10–12.

(3)   Page 7, line 24, "Bac" -> "BAC"

R: Thanks for pointing out the unsuitable expression. We have corrected the mistake accordingly. Please refer to Page 7 Line 28, Page 7 Line 30, and Page 8 Line 3.

(4) Page 9, line 17, why do the authors think "it suggests a strong sytematic coupliing between ML and FT, ...", instead of they are both modulated by the same mechanism, like regional transport of aerosols.

R: Thanks for pointing out the unsuitable expression. It is possible that they are both modulated by the same mechanism, which was overlooked in our previous analysis, and we have corrected our statement: "They may be regulated by the same mechanisms, such as regional transport of aerosols.". Please refer to Page 9 Line 28–29 and Page 18 Line 26.

(5) Page 10, line 25, PDR at 532 should be at percentage, namely, 0.082, or adding a percent sign (%) instead. And the authors need to check the manuscript thoroughly, because there are many places with this error.

R: Thank you for your comments, we have corrected the relevant mistakes and carefully checked the manuscript for similar errors.

(6) Page 11, line 33, "building warming" -> "building heating".

R: Thanks for pointing out the unsuitable expression. We have followed this suggestion and corrected it accordingly. Please refer to Page 12 Line 13.

(7) Page 11, line 33, "MEGGA" -> "MERRA". (also in caption of fig. 8)

R: We have followed this suggestion and corrected the mistake accordingly. Please refer to Page 12 Line 13.

(8) Page 13, line 17, "upper air pollution transport" is more appropriate.

R: Thanks for pointing out the unsuitable expression. We have followed this suggestion and corrected it accordingly. Please refer to Page 14 Line 5.

(9)    Page 15, line 22, "bottom" should be removed.

R: Thanks for pointing out the unsuitable expression. We have followed this suggestion and corrected it accordingly.

(10)    Page 15, line 29-32, correlation of coefficient cannot be used to determine the goodness of fit for non-linear models. Therefore, it cannot be compared between linear fitting and non-linear fitting, just by looking at correlation of coefficient. The author should either use a different metric to do the comparison or remove such statement.

R: Thanks for pointing out the unsuitable expression. We have removed these statements in the manuscript.

(11)    Page 17, line 14-15, the authors should be specific when mentioning "near the ML" or "around ML" (the same page, line 18).

R: Thank you for your comments, here we discussed the relationship between PRL derived MLH and vertical profiles of anthropogenic aerosols and dust. The high dust mass concentration usually distributed near the ML (1.4-3.4 km) during the whole observation period, and high dust mass concentration usually distributed near the ML (1.6-3.4 km) in spring. We have specified the "near the ML" and "around ML" in the manuscript. Please refer to Page 18 Line 3–7.

(12)    Page 18, line 20-21, the authors should clarify why "the bottom dust mass concentration is mainly influenced by transport" instead of by local sources.

R: Thank you for your comments. Our observation site is located in the urban area of Beijing, where soil dust, construction dust, coal dust and motor vehicle exhaust are the four main sources of dust in Beijing (Wang et al., 2015), accounting for 38.50%, 22.25%, 14.06%, and 20.82% of the total amount of dust, respectively. The bottom dust mass concentration discussed in the manuscript is located at 0.25 km, while soil dust, construction dust, coal dust and motor vehicle exhaust usually concentrated within tens of meters (Noh et al., 2021). Thus, the bottom dust mass concentration is mainly influenced by transport, and we have added these statements and specified "bottom

(0.25 km)" in the manuscript. Please refer to Page 17 Line 2–4 and Page 19 Line 9.

(13) In caption of fig 2, Check about the conversion factors of dust and anthropogenic aerosols. It's too low for dust and a little high for anthropogenic aerosols (see ref.[1-2]).

R: Thanks for your suggestion, we also found this phenomenon when retrieving the dust and anthropogenic aerosol mass concentrations. We have carefully checked the sun–photometer derived aerosol optical parameters and found no problem. This phenomenon is very interesting, and it is worth further exploring why the conversion factors of dust in Beijing is low, and the conversion factors of anthropogenic aerosol aerosols is high.

**Reference:**

Tang, G., Zhu, X., Hu, B., Xin, J., Wang, L., Münkel, C., Mao, G., and Wang, Y.: Impact of emission controls on air quality in Beijing during APEC 2014: lidar ceilometer observations, Atmospheric Chemistry and Physics, 15, 12667-12680, 10.5194/acp-15-12667-2015, 2015.

Tang, G., Zhang, J., Zhu, X., Song, T., Münkel, C., Hu, B., Schäfer, K., Liu, Z., Zhang, J., Wang, L., Xin, J., Suppan, P., and Wang, Y.: Mixing layer height and its implications for air pollution over Beijing, China, Atmospheric Chemistry and Physics, 16, 2459-2475, 10.5194/acp-16-2459-2016, 2016.

Wu T, Li Z, Chen J, et al. Hygroscopicity of different types of aerosol particles: Case studies using multi-instrument data in megacity Beijing, China[J]. Remote Sensing, 2020, 12(5): 785.

Xia C, Sun J, Qi X, et al. Observational study of aerosol hygroscopic growth on scattering coefficient in Beijing: A case study in March of 2018[J]. Science of the Total Environment, 2019, 685: 239-247.

Dawson K W, Ferrare R A, Moore R H, et al. Ambient aerosol hygroscopic growth from combined Raman lidar and HSRL[J]. Journal of Geophysical Research: Atmospheres, 2020, 125(7): e2019JD031708.

Sicard, M., Pérez, C., Rocadenbosch, F., Baldasano, J., and García-Vizcaino, D. J. B.-

L. M.: Mixed-layer depth determination in the Barcelona coastal area from regular lidar measurements: methods, results and limitations, 119, 135-157, 2006.

Flamant, C., Pelon, J., Flamant, P. H., and Durand, P. J. B.-L. M.: Lidar determination of the entrainment zone thickness at the top of the unstable marine atmospheric boundary layer, 83, 247-284, 1997.

Wang R, Zou X, Cheng H, et al. Spatial distribution and source apportionment of atmospheric dust fall at Beijing during spring of 2008–2009[J]. Environmental Science and Pollution Research, 2015, 22: 3547-3557.

Noh S Y, Ji J H, Kim H S, et al. Required flux tower height for measurement of re-suspended road dust[J]. Journal of Mechanical Science and Technology, 2021, 35: 1781-1789.

Mamouri R, Ansmann A. Fine and coarse dust separation with polarization lidar. Atmospheric Measurement Techniques. 2014;7(11):3717-35.

Ansmann A, Mamouri R-E, Hofer J, Baars H, Althausen D, Abdullaev SF. Dust mass, cloud condensation nuclei, and ice-nucleating particle profiling with polarization lidar: updated POLIPHON conversion factors from global AERONET analysis. Atmospheric Measurement Techniques. 2019;12(9):4849-65.

**Response to reviewer 2:**

We truly grateful for the reviewers' positive assessments of our manuscript and the helpful suggestions. We have revised the manuscript carefully according to the reviewers' comments. Point-to-point responses are given below. The original comments are black in color, while our responses are in blue. The revised parts in the manuscript are marked in red. All the page number and line number are referred to the revised manuscript.

**Major comments**

(1)  There are many studies on air pollution in Beijing in the past decade, and the authors need to highlight the innovations of this paper to distinguish it from other studies. In other words, this study revealed the vertical distribution characteristics of anthropogenic aerosols and dust mass concentrations, what kind of research can be conducted in the future based on polarization Raman dataset presented in the manuscript. What is the significance or implications of those research results for air pollution in Beijing.

R: We think this comment is meaningful and valuable. We also mentioned in the article that most of the previous studies focused on the surface air pollution in Beijing, and mainly focused on the total aerosol mass concentration. In this study, the vertical distribution characteristics of aerosol types and their optical properties were captured by long−term continuous polarization Raman lidar observation. We focused on the long–term vertical distributions of dust (coarse) and anthropogenic aerosols (fine) and their relationships with mixing layer height, which have not been revealed in previous studies, and found large amounts of anthropogenic aerosols accumulate at the top of the mixing layer, which is most noticeable in summer. This also provides a new impetus for research on the relationship between vertical distribution of air pollutants and mixing layer height. In addition, the data set presented in the manuscript can also be used for further research, for example:

1.  At the end of 2019 and the beginning of 2020, the epidemic control policy led to the reduction of anthropogenic emissions in Beijing. The effect of anthropogenic

emission reduction on the vertical structure of fine particulate matter in Beijing can be further discussed.

2. We found a large number of fine particles at the top of the mixed layer in Beijing, while the formation, accumulation and dissipation mechanism of air pollutants at the top of the mixing layer, as well as its impact on the surface radiation feedback, still need to be investigated in detail. Although we speculate that the aerosol hygroscopic growth may be a key process, there is no direct evidence.

3. Our observation results, especially the vertical profile of different aerosol types, can also be integrated into the dust generation and convection and chemical migration models of the North China Plain. Lidar data assimilation has long been recognized for its potential to improve numerical modeling analyses (Zhang et al., 2011). Recent research presents the assimilation of CALIPSO extinction coefficient measurements in the chemistry transport model, they focus on the dessert dust outbreak and found that the assimilation of CALIPSO lidar observations improves the statistics compared to the model free run (Amraoui et al., 2020). In addition, the vertical profile of depolarization ratio and lidar ratio, as well as the aerosol classification results can be used as the assimilation data of the model to optimize the simulation. With integrated information from various sources, i.e. numerical simulation, ground-based and satellite remote sensing, these results can more accurately describe the three-dimensional distribution pattern of aerosols.

4. Our data can also be used to support basic data analysis for spaceborne lidar missions such as CALIPSO (Cloud–Aerosol Lidar and Infrared Pathfinder Satellite Observations) (Winker et al., 2009), ALADIN (Atmospheric LAser Doppler INstrument) (Witschas et al., 2020), and ATLID (Atmospheric Lidar) (Illingworth et al., 2015), upgrading the accuracy of regional terrestrial and global satellite lidar inversions.

We also include these important perspectives in the manuscript. Please refer to Page 3 Line 28–31 and Page 19 Line 16–26 in the manuscript.

**Other comments**

(1)  In section 2.2-2.5, the author uses a variety of data. It is necessary to introduce the main purpose of the data and its role in this article before starting.

R: Thanks for your suggestion, we have added an introduction to the data and methods before section 2.1. "In this section, we specify the instrument, materials and methods employed throughout the study. Section 2.1 introduces the PRL system, Section 2.2-2.5 describes the auxiliary data, and the polarization lidar photometer networking (POLIPHON) method for retrieving dust and anthropogenic aerosols mass concentration is described in Section 2.6. Aerosol Robotic Network (AERONET) aerosol optical parameters were the input parameters of POLIPHON method, surface $PM_{10}$ and $PM_{2.5}$ mass concentrations are used for validation of POLIPHON method results, Hybrid Single Particle Lagrangian Integrated Trajectory (HYSPLIT) model and reanalysis data are used for auxiliary analysis." Please refer to Page 3 Line 5–10 in the manuscript.

(2)  In section 3, the data was from 22 May 2019 to 20 February 2022. But PRL data were missing from 16 November 2020 to 29 May 2021. In the following Fig.3, Fig.5, Fig.7, Fig.9 and Fig.10, the monthly average data was used. It is necessary to clarify which period of time is used for averaging.

R: We discuss the coverage of PRL measurements in detail in Section 2.1 and present the relevant content in the supplemental materials. we excluded signals when the signal to noise ratio was less than 1, as well as data measured under rain, snow and low cloud conditions. Percentage of analyzed PRL measurements from May 2019 to February 2022 are shown in Figure R1, the unanalyzed measurements of "Shutdown" and "Rain or Cloud" are due to the instrument failure or weather conditions.

[Figure]

***Figure R1****. Coverage of Polarized Raman Lidar (PRL) measurements from May 2019 to February 2022. (a) Percentage of analyzed PRL measurements, the unanalyzed measurements of "Shutdown" and "Rain or Cloud" are due to instrument failure or weather conditions. The analyzed number of (b) EXT$_{532}$, (c) PDR$_{532}$, and LR$_{355}$ points at different heights. The PRL detection error increases with the increase of height, and the number of points also decreases with the increase of height.*

(3) Page 9, line 8, "Fig1b shows the PRL-derived AOD in the ML and FT", Fig1b? It should be Fig. 3b.

R: Thanks for pointing out the unsuitable expression. We have corrected it accordingly. Please refer to Page 9 Line 19 in the manuscript.

(4) Page 11, line 32, "MERRA" is misspelled.

R: Thanks for pointing out the unsuitable expression. We have followed this suggestion and corrected it accordingly. Please refer to Page 12 Line 13 in the manuscript.

(5) Page 14, line 19, "from the surface to more than 5 km". Data below 0.25 km and above 5 km are not presented in the manuscript.

R: Thank you for your comments. We don't show it in the figure, but the data set we provide shows dust aerosols over 5 km. The surface $PM_{10}$ mass concentration also captures dust aerosols. Thus, the distribution of Asian dust can extend from the surface up to more than 5 kilometers.

(6) Page 16, line 31, explain "non-dust", is it anthropogenic aerosols?

R: Here we refer to anthropogenic aerosols, and we have corrected it in the manuscript. Please refer to Page 17 Line 19 in the manuscript.

(7) Page 17, line 2, "dust aerosol (about 50 µg/m3) is mainly above the ML". Not only above the mixed layer, the dust concentration is also higher below mixing layer.

R: The dust concentration is indeed higher below the mixing layer, but we want to emphasize that the dust is mainly concentrated above the mixed layer. Obviously, the dust mass concentration above the mixed layer is higher than that in the mixed layer.

(8) In section 3.3.2, the concentration of anthropogenic aerosols was low in winter. However, during the observation period, especially at the end of 2019 and the beginning of 2020, epidemic control policy led to the reduction of anthropogenic emissions. Whether these reductions had any effect on the observations of this study? and these effects also remain to be explored.

R: Obviously, reductions in anthropogenic emissions can affect our observations. There have also been many studies analyzing the impact of lockdown during the epidemic on air pollution in Beijing (Hu et al., 2021, Zhao et al., 2020, Zhang et al., 2022). Hu et al. (Hu et al., 2021) found that the epidemic control policy led to the reduction of air pollutants in Beijing during the 2020 Spring Festival by 35.1%-51.8%. However, the meteorological conditions during the Spring Festival in 2020 are not conducive to the diffusion of air pollutants, leading to the occurrence of haze episodes. During 23-28 January and 8-13 February, Beijing experienced two large-scale air pollution events. The first is mainly affected by local emissions, such as building heating, and the second is mainly affected by regional transport over the North China Plain. The observation of

PRL also captures the vertical distribution of air pollutants during the COVID-19 period, but this manuscript mainly discusses the long-term evolution of air pollutants and their relationships with mixing layer height, and these effects can be studied in the future.

(9) In discussion section, authors found that there is a significant negative correlation between anthropogenic aerosols and MLH in four seasons. Are there any similar observation results or simulation results with the same conclusions as this paper?

R: There are plenty of theoretical and observational studies on aerosols in Beijing (Zhong et al., 2019; Guo et al., 2016; Miao et al., 2015; Miao et al., 2018), most of them only consider total aerosol mass concentrations and focus on heavy pollution episodes, the long–term evolution of dust (coarse) and anthropogenic aerosols (fine) and their relationships with mixing layer height have not been revealed, which is also one of the innovations of this paper.

(10) In section 5, the conclusion should point out the shortcomings in this study and future research perspectives.

R: This is a very valuable comment. Although our results elucidate the long–term vertical distributions of dust (coarse) and anthropogenic aerosols (fine) and their relationships with mixing layer height, our research also has two shortcomings. Firstly, PRL has incomplete overlap region, about 0.25 km, which prevents us from capturing the evolution of air pollutants at the lowest level (0-0.25 km). Due to the incomplete overlap region, our inversion of the MLH also starts from 0.25 km, which may lead to the overestimation of the MLH. Secondly, due to the limitations of the POLIPHON method, we excluded cases with relative humidity greater than 85%, and the accumulation of particles at the top of the ML may undergo a significant hygroscopic growth, so the anthropogenic aerosols mass concentration at the top of the ML may be underestimated. The perspectives of the research are detailed in "Major comments".

We also include these shortcomings and perspectives in the manuscript. Please refer to Page 19 Line 16–26 in the manuscript.

(11) Suggestion: It would be better to combine Figure 11 and Figure 12 into a single figure, and also for Figure 13 and Figure 14.

R: We have followed this suggestion and combine Figure 11 and Figure 12 into a single figure, and also for Figure 13 and Figure 14. Please refer to Fig.12 and Fig.13 in the manuscript.

**Reference:**

Zhang, J., J.R. Campbell, J. S. Reid, D. L. Westphal, N. L. Baker, W. F. Campbell, and E. J. Hyer, Evaluating the impact of assimilating CALIOP-derived aerosol extinction profiles on a global mass trans-port model, Geophys. Res. Lett., 38, L14801, doi:10.1029/2011GL047737, 2011.

El Amraoui, L., Sič, B., Piacentini, A., Marécal, V., Frebourg, N., and Attié, J.-L.: Aerosol data assimilation in the MOCAGE chemical transport model during the TRAQA/ChArMEx campaign: lidar observations, Atmos. Meas. Tech., 13, 4645–4667, https://doi.org/10.5194/amt-13-4645-2020, 2020.

Winker, D. M., Vaughan, M. A., Omar, A., Hu, Y., Powell, K. A., Liu, Z., Hunt, W. H., and Young, S. A.: Overview of the CALIPSO Mission and CALIOP Data Processing Algorithms, JAtOT, 26, 2310-2323, 10.1175/2009jtecha1281.1, 2009.

Witschas, B., Lemmerz, C., Geiß, A., Lux, O., Marksteiner, U., Rahm, S., Reitebuch, O., and Weiler, F. J. A. M. T.: First validation of Aeolus wind observations by airborne Doppler wind lidar measurements, 13, 2381-2396, 2020.

Illingworth, A. J., Barker, H., Beljaars, A., Ceccaldi, M., Chepfer, H., Clerbaux, N., Cole, J., Delanoë, J., Domenech, C., and Donovan, D. P. J. B. o. t. A. M. S.: The EarthCARE satellite: The next step forward in global measurements of clouds, aerosols, precipitation, and radiation, 96, 1311-1332, 2015.

Zhao N, Wang G, Li G, et al. Air pollution episodes during the COVID-19 outbreak in the Beijing–Tianjin–Hebei region of China: an insight into the transport pathways and source distribution[J]. Environmental Pollution, 2020, 267: 115617.

Hu J, Pan Y, He Y, et al. Changes in air pollutants during the COVID-19 lockdown in Beijing: insights from a machine-learning technique and implications for future

control policy[J]. Atmospheric and Oceanic Science Letters, 2021, 14(4): 100060.

Zhang H, Wang X, Shen X, et al. Analysis of air pollution characteristics, transport pathways and potential source areas identification in Beijing before, during and after the COVID-19 outbreak[J]. Frontiers in Environmental Science, 2022, 10: 982566.

Zhong, J., Zhang, X., Wang, Y., Wang, J., Shen, X., Zhang, H., Wang, T., Xie, Z., Liu, C., Zhang, H., Zhao, T., Sun, J., Fan, S., Gao, Z., Li, Y., and Wang, L.: The two-way feedback mechanism between unfavorable meteorological conditions and cumulative aerosol pollution in various haze regions of China, Atmospheric Chemistry and Physics, 19, 3287-3306, 10.5194/acp-19-3287-2019, 2019.

Guo, J., Miao, Y., Zhang, Y., Liu, H., Li, Z., Zhang, W., He, J., Lou, M., Yan, Y., Bian, L., and Zhai, P.: The climatology of planetary boundary layer height in China derived from radiosonde and reanalysis data, Atmospheric Chemistry and Physics, 16, 13309-13319, 10.5194/acp-16-13309-2016, 2016.

Miao, Y., Liu, S., Guo, J., Huang, S., Yan, Y., and Lou, M.: Unraveling the relationships between boundary layer height and PM(2.5) pollution in China based on four-year radiosonde measurements, Environ Pollut, 243, 1186-1195, 10.1016/j.envpol.2018.09.070, 2018.

Miao, Y., Hu, X. M., Liu, S., Qian, T., Xue, M., Zheng, Y., and Wang, S. J. J. o. A. i. M. E. S.: Seasonal variation of local atmospheric circulations and boundary layer structure in the Beijing-Tianjin-Hebei region and implications for air quality, 7, 1602-1626, 2015.